# Creating two self-assembly micro-environments to achieve supercrystals with dual structures using polyhedral nanoparticles

Yih Hong Lee [1], Chee Leng Lay[1,2], Wenxiong Shi [3], Hiang Kwee Lee[1,2], Yijie Yang[1], Shuzhou Li [3] & Xing Yi Ling [1]

Organizing nanoparticles into supercrystals comprising multiple structures remains challenging. Here, we achieve one assembly with dual structures for Ag polyhedral building blocks, comprising truncated cubes, cuboctahedra, truncated octahedra, and octahedra. We create two micro-environments in a solvent evaporation-driven assembly system: one at the drying front and one at the air/water interface. Dynamic solvent flow concentrates the polyhedra at the drying front, generating hard particle behaviors and leading to morphology-dependent densest-packed bulk supercrystals. In addition, monolayers of nanoparticles adsorb at the air/liquid interface to minimize the air/liquid interfacial energy. Subsequent solvent evaporation gives rise to various structurally diverse dual-structure supercrystals. The topmost monolayers feature distinct open crystal structures with significantly lower packing densities than their densest-packed supercrystals. We further highlight a 3.3-fold synergistic enhancement of surface-enhanced Raman scattering efficiency arising from these dual-structure supercrystals as compared to a uniform one.

[1] Division of Chemistry and Biological Chemistry, School of Physical and Mathematical Sciences, Nanyang Technological University, 21 Nanyang Link, Singapore 637371, Singapore. [2] Agency for Science, Technology and Research (A*STAR), Institute of Materials Research and Engineering, 2 Fusionopolis Way, Innovis, #08-03, Singapore 138634, Singapore. [3] School of Materials Science and Engineering, Nanyang Technological University, 50 Nanyang Avenue, Singapore 639798, Singapore. Correspondence and requests for materials should be addressed to X.Y.L.(email: xyling@ntu.edu.sg)

Nature is replete with sophisticated structures that are important for survival: a well-known example is the dual crystal structures beneath a chameleon's skin that enable dynamic camouflaging and body temperature modulation[1]. Drawing inspiration from nature, we hypothesize that achieving large-area supercrystals with multiple structures can serve as an attractive approach to enhance material properties. This hypothesis is built upon the well-known dependence of material properties on crystal structures[2–5], and is manifested most evidently across various optical[6–10] and electrical[11–13] studies. However, the lack of large-area supercrystals with more than one structure implies a scarcity of insight on their potential property. Currently, the major experimental challenge is to assemble nanoparticles into supercrystals containing more than one crystal structure. Supercrystals with more than one structure are elusive because nanoparticles are typically assembled in a homogeneous environment, in which nanoscale control over their organization has led to numerous morphology-dependent supercrystals[14–25], binary nanocrystal superlattices[26–28], mesophases[29–35], as well as quasicrystals[36–38]. To date, there has only been the observation of interfacial boundaries between crystal grains[36,39]. The locations of these boundaries are arbitrary among the supercrystal, thus precluding their scalable assembly into macroscopic structures.

Herein, we demonstrate the concept of one assembly with dual structures for a family of shape-controlled Ag polyhedra. Our building blocks comprise two Platonic (cubes and octahedra) and three Archimedean (truncated cubes, cuboctahedra, and truncated octahedra) solids. Aside from nanocubes, these polyhedra spontaneously organize into large-area structurally diverse densest-packed supercrystals as well as open structures with significantly lower packing densities. We create two self-assembly micro-environments in the solvent evaporation-driven self-assembly of these Ag polyhedra: one at the air/liquid interface and another at the drying front within the droplet of nanoparticle dispersion. Monolayers of nanoparticles adsorb at the air/liquid interface to minimize interfacial energy. At the same time, dynamic particle transport arising from convective solvent flow during evaporation creates a non-equilibrium distribution of Ag polyhedra within the droplet. Nanoparticles accumulate at the drying front and adopt hard particle behaviors, in turn generating morphology-dependent densest-packed bulk supercrystals. Solvent evaporation results in the formation of large-area morphology-dependent open structures on the topmost layers in conjunction with densest-packed supercrystals in the bulk. These topmost layers exhibit morphology-dependent reduced translational or orientational orders. We further demonstrate that such assembled dual structures boost the overall surface-enhanced Raman scattering (SERS) efficiency as compared to a supercrystal with only one crystal structure.

## Results

### Dual-structure supercrystal assembled using Ag octahedra.
Single crystalline Ag polyhedra in our experiments are synthesized via the polyol reduction route[40], and are passivated with a layer of poly(vinylpyrrolidone) (PVP). Nanocubes and octahedra have their square [100] and triangular [111] facets exposed, respectively; truncating the nanocubes introduces triangular [111] facets to the eight nanocube vertices, eventually leading to the formation of the cuboctahedra with eight [111] and six [100] facets (Fig. 1a). To create dual self-assembly micro-environments, a 10 μL dispersion of Ag nanoparticles is drop cast onto a Si substrate and is allowed to dry (Fig. 1b). The Si substrate is then cracked open for top-view and cross-sectional scanning electron microscopy (SEM) characterization.

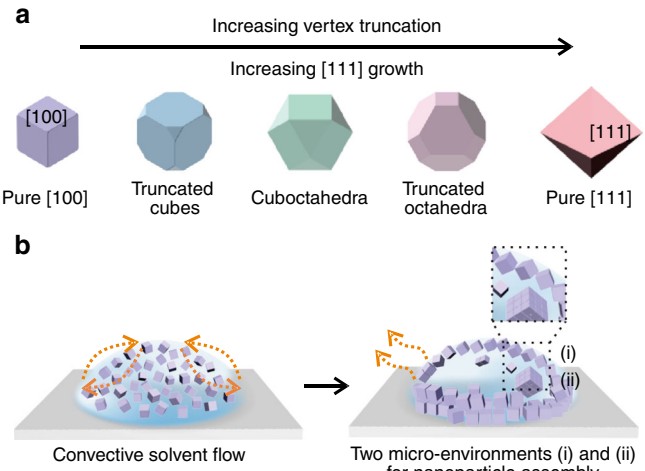

**Fig. 1** Dual micro-environments to achieve one assembly with two structures. **a** Ag polyhedra building blocks range from cubes to octahedra. **b** A droplet of nanoparticle dispersion is allowed to dry on a substrate. Convective solvent flow within the droplet during solvent evaporation concentrates Ag polyhedra near the drying front, giving rise to two unique regions of crystallization (i and ii)

Two types of assembled structures are clearly discernible from the cross-sectional SEM image of the supercrystal assembled using octahedra as the building blocks (Fig. 2). The center portion shows large-area crystalline grains of octahedra stretching up to 0.5 mm from the drying front when water is used as the solvent (Fig. 2a, b, Supplementary Fig. 1). The supercrystal here exhibits both long-range translational and orientational order. Fast-Fourier transform (FFT) analysis of the supercrystal in the center region indexes it to the densest-packed Minkowski lattice (Fig. 2e, f). The Minkowski lattice has a primitive triclinic unit cell with a high packing efficiency of ~95% (Supplementary Fig. 2).

In contrast, octahedra in the topmost layer organize into a more open structure with reduced translational order (Fig. 2b–d, blue colored layer). Octahedra in this layer balance on the Minkowski lattice below, with the triangular [111] facets of the top layer octahedra fitting into interstices (Fig. 2c). Top-view characterization of this layer shows a loosely packed lattice, with clear gaps between neighboring rows of octahedra (Fig. 2d, Supplementary Fig. 3). FFT analysis of the topmost layer demonstrates the crystallinity of this structure (inset of Fig. 2d), with the experimentally determined packing efficiency estimated to be 49% (Supplementary Fig. 4, Supplementary Note 1). A displacement in the lattice positions of the octahedra in this topmost layer implies a decrease in translational order relative to the bulk Minkowski lattice. Indeed, order analyses performed using radial distribution function show broader, less distinct, and lower-intensity peaks with increasing particle separation for the topmost octahedra layer, affirming decreased translational order of this layer relative to the Minkowski lattice (Fig. 2g). On the other hand, orientational order is evident for this top layer, with octahedra oriented along the same direction. Orientational order analysis performed using the deuterium order parameter indicates that there is negligible reduction of orientational order between the topmost layer and the Minkowski lattice (Supplementary Table 1, Supplementary Discussion 1). Our experimental observations here highlight the formation of two distinct structures within a single assembly of Ag octahedra, with the reduced translational order of the topmost layer resembling a nematic liquid crystalline phase.

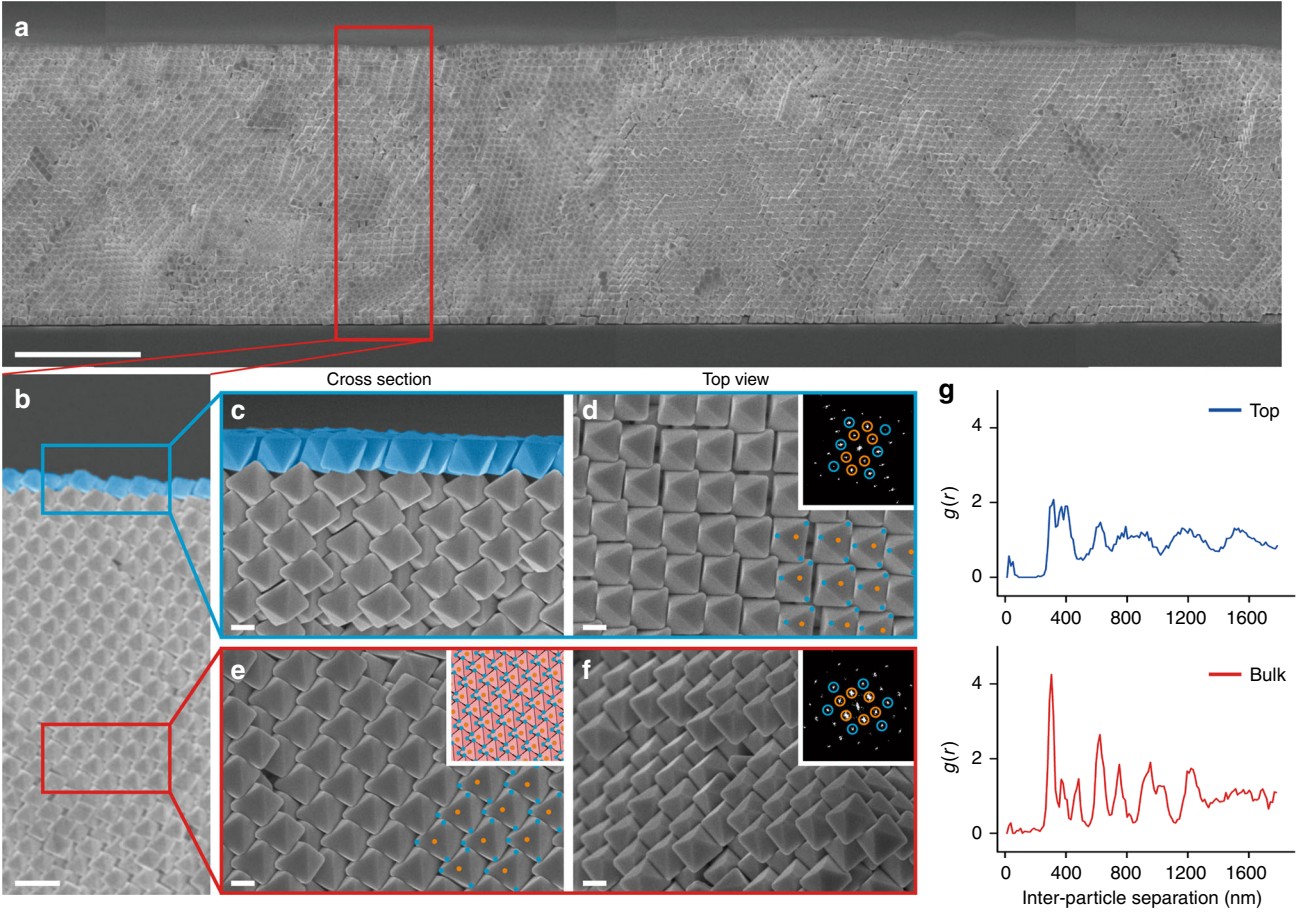

**Fig. 2** Dual crystal structure formed using Ag octahedra in water. **a**, **b** Cross-sectional panoramic characterization shows the formation of large-area supercrystals with long-range order. The heights of the supercrystal are nearly uniform at ~10 μm and stretching more than 50 μm across. **c–f** The supercrystal can be divided into two distinct regions. **c**, **d** The top layer features an open structure with Ag octahedra possessing long-range orientational order and reduced translational order relative to the bulk. **e**, **f** The bulk of the supercrystal is crystalline, and can be indexed to the Minkowski lattice (inset of **e**). **a** Scale bar, 5 μm; **b** scale bar, 1 μm; **c–f** scale bars, 200 nm. Insets of **d**, **f** are the FFT images of the respective crystal structure. The Ag octahedra in the open structure resembling a liquid crystalline phase are color-coded for ease of discussion. **g** Radial distribution functions of the respective crystals

**Morphology-dependent open structures and bulk supercrystals.** In addition, structurally diverse open structures with reduced orientational order on the topmost layer can be achieved as the building block morphology changes from nanocubes to truncated nanocubes, to cuboctahedra, and to truncated octahedra (Fig. 3, Supplementary Fig. 5). Long-range translational order of the structures is evident from the SEM, FFT, and RDF profiles for all the open structures observed (Fig. 3a–h, Supplementary Fig. 6). Nanocubes and truncated nanocubes form a hexagonal structure with majority of the nanocubes standing on their [111] vertices in the topmost layer, resembling the non-close-packed rhombille tiling patterns seen in Escher's works (Fig. 3a, b, e, f, m). This hexagonal structure is distinct from the more commonly observed thermodynamically stable square lattice of nanocubes. Cuboctahedra adopt a square network with individual particles regularly spaced apart from each other akin to an egg tray (Fig. 3c, g). A larger population of the cuboctahedra have the triangular [111] facets facing up than the square [100] facets (Fig. 3m). Truncated octahedra form a hexagonal network with majority of the particles having their [110] facets facing up and a smaller population of hexagonal [111] facets facing up (Fig. 3d, h, m). Orientational order analyses performed using the deuterium order parameter further affirms reduced orientational order among these topmost layers (Supplementary Table 1, Supplementary Discussion 1).

For truncated nanocubes, cuboctahedra, and truncated octahedra, our collective experimental observations highlight a resemblance to a plastic crystalline phase, which are less frequently observed in assembled structures using nanoparticle building blocks.

The open structures observed in our experiments are again distinct from their corresponding bulk supercrystals. In the bulk, Ag polyhedra organize into their densest-packed lattices (Fig. 3i±l, Supplementary Fig. 7). Cross-sectional SEM characterization show that nanocubes and truncated nanocubes assemble into a tilted simple cubic supercrystal (Fig. 3i, j). Both nanocubes and truncated nanocubes contact each other face-to-face in this supercrystal, and are different from the edge-to-edge contact mode in the hexagonal array in the topmost layer. Cuboctahedra form a tilted body-centered tetragonal lattice (Fig. 3k), and truncated octahedra assemble into a body-centered cubic lattice (Fig. 3l). The differences observed from the FFT patterns, RDF profiles, as well as close-up SEM images for the open structures in the topmost layers and the bulk supercrystals further show that they are two distinct structures (Supplementary Fig. 8). We note that this distinction manifests more evidently in truncated nanocubes, cuboctahedra and truncated octahedra than in nanocubes (vide infra).

Furthermore, the bulk supercrystals possess higher packing densities than the open structures in the topmost layer.

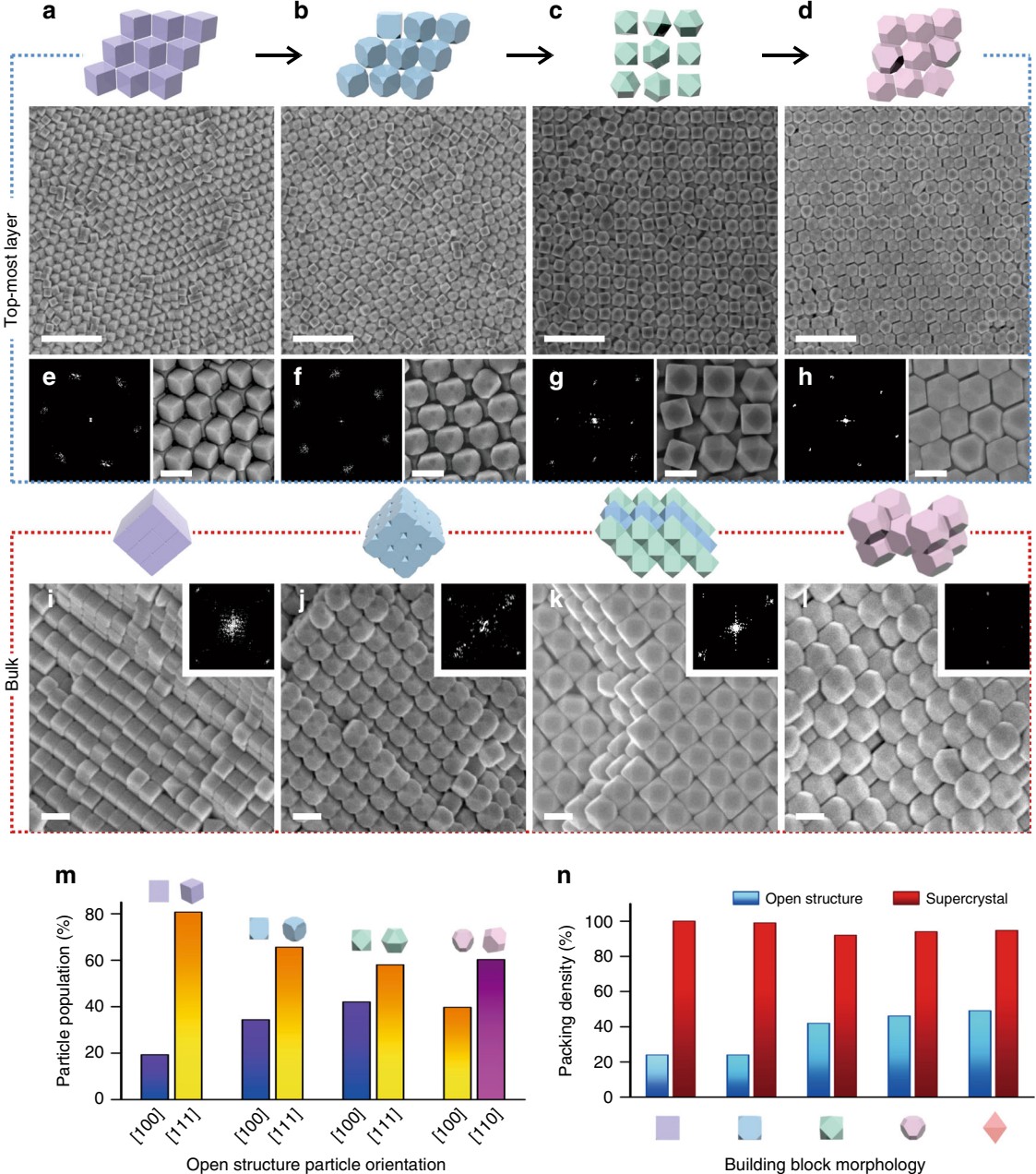

**Fig. 3** Dual crystal structures of Ag polyhedra. Top-view SEM characterization of **a** nanocubes, **b** truncated nanocubes, **c** cuboctahedra, and **d** truncated octahedra. **e**–**h** FFTs and close-up images of the various assembled structures. **i**–**l** Cross-sectional characterization of the bulk supercrystals. Insets are FFTs of the bulk crystal structures. **a**–**d** Scale bars, 1 μm; **e**–**l** scale bars, 200 nm. **m** Relative population of the various orientations adopted by the open structures of Ag polyhedra. **n** Packing densities of the various open structures as well as their supercrystals

Nanocubes are space-filling with a packing efficiency of 100% while that of truncated nanocubes is nearly 100% due to slight vertex truncation (Fig. 3n). On the other hand, the hexagonal structure of both the nanocubes and truncated nanocubes has a significantly lower packing density of ~24%[7] (Fig. 3n). The supercrystal of cuboctahedra has a packing density of ~92%[41], whereas its open square network in the topmost layer has a packing density of ~42% (Supplementary Fig. 9, Supplementary Note 2). Truncated octahedra supercrystal has a packing efficiency of ~94%[41], while its hexagonal open structure has an approximate packing efficiency of 46% (Supplementary Fig. 10, Supplementary Note 3).

As the first experimental observation of morphology-dependent open structures resembling liquid/plastic crystals for

the various polyhedral building blocks, we qualify this resemblance through standard order analyses methods using both RDF and deuterium order parameter. As implied in earlier discussion, the former evaluates translational order while the latter examines orientational order. We compare both order parameters between the topmost layer and the bulk supercrystal to determine whether there is a decrease in either order for the topmost layer. The underlying argument for this comparison is that as a crystalline solid, the densest-packed bulk supercrystal should exhibit the highest degree of translational and orientational order. A decrease in translational (orientational) order relative to the bulk thus suggests a resemblance to liquid (plastic) crystals in more conventional systems. In fact, various instances of nanoparticle-based liquid crystalline structures with good translational order

have been demonstrated using nanorods[29] and bipyramids[34] as building blocks.

**Air/liquid interface drives open structure formation.** Our experimental observations collectively highlight the presence of two self-assembly micro-environments within the drying nanoparticle dispersion which give rise to the two crystal structures observed across the various polyhedral building blocks. The first region corresponds to the air/water interface, which is an important feature for open structure formation on the topmost layer (Fig. 2b). The second region corresponds to the bulk phase near the drying front. Using Ag octahedra as the model study system, the open structures resembling liquid crystals persistently form on the topmost layer even when the droplet-laden substrate is inverted or tilted during the drying process (Fig. 4a). Both experimental setups eliminate the possibility of gravity in driving the formation of the topmost open structure. In addition, varying the duration of solvent evaporation between an hour to ~3 days also does not lead to a switch of the topmost liquid crystalline phase to the Minkowski lattice (Supplementary Fig. 11). Furthermore, similar open structures are observed when the self-assembly solvent is changed to N,N-dimethylformamide (DMF), 1-propanol, and 1-hexanol (Fig. 4b, Supplementary Fig. 12). On the other hand, previous self-assembly experiments based on the sedimentation of Ag octahedra in the absence of an air/liquid interface did not give rise to structures resembling liquid crystalline phases[14], further affirming the role of the interface in driving liquid crystal formation.

Minimizing the air/liquid interfacial energy drives the adsorption of a monolayer of particles in our experiments, and has direct implications for open structure formation in our experiments. Dynamic surface tension measurements performed using the pendant drop method show that the aqueous particle-laden droplet has a lower initial value of 64 mN m$^{-1}$ as compared to the 72 mN m$^{-1}$ for a pure water droplet. A continual decrease in surface tension with time is also observed for the particle-laden droplet, decreasing from ~64 to ~55 mN m$^{-1}$ over the course of measurements (Fig. 4c). In contrast, the surface tension of a pure water droplet remains constant over the same period of measurement. The related change in interfacial energy is estimated to be approximately $-10^6$ $k_BT$, based on the equation $\Delta E = (\gamma_0 - \gamma)\pi R^2/\eta$[42]. The change in interfacial energy ($\Delta E$) is dependent on the nanoparticle radius ($R$) and its packing fraction ($\eta$); $\gamma_0$ and $\gamma$ refer to the initial and final interfacial tension measured. This energy is significantly larger than both thermal fluctuations and gravity, which are typically several $k_BT$. Furthermore, we record a consistent decrease of ~9% in surface tension for the particle-laden droplets in various solvents as compared to the pure solvents in our static surface tension measurements (Fig. 4d, Supplementary Table 2). Notably, the decrease in static surface tension is similar across a wide range of solvents tested. These findings imply the role of the air/liquid interface as a unique micro-environment to drive a similar process of Ag octahedra adsorption onto the droplet's surface across various solvents (Fig. 4e).

Moreover, additional control experiments to visualize the in situ orientation[43] of Ag octahedra and nanocubes show the formation of semi-crystalline structures at the air/liquid interface (Supplementary Figs. 13, 14). Notably, the dominant nanoparticle orientations on the topmost layers correspond to the nanoparticle vertices or edges (Fig. 3m, Supplementary Figs. 13, 14), and are regions of relatively larger curvatures as compared to the facets for the respective nanoparticle morphologies. These regions have lower densities of PVP as compared to the other parts of the nanoparticles, similar to other systems such as oleic

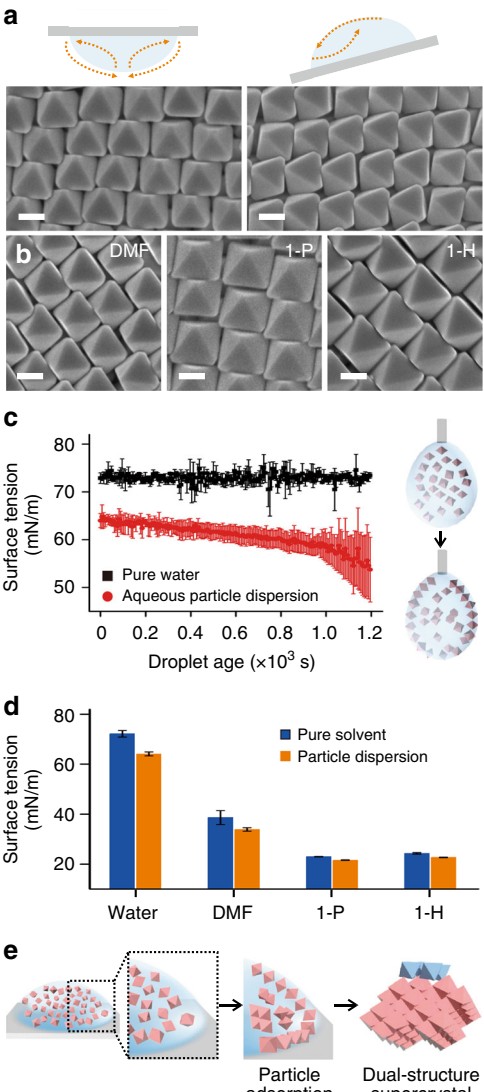

**Fig. 4** Air/liquid interface drives open structure formation. **a** Open structure persists when the droplet dries in an upside-down configuration or at a tilt angle. Schematics illustrate assembly setups, arrows indicate direction of solvent flow. **b** Open structure formation in N,N-dimethylformamide (DMF), 1-propanol (1-P), and 1-hexanol (1-H). All scale bars, 200 nm. **c** Time-dependent surface tension measurements of an aqueous particle-loaded pendant drop (red line) and a pure solvent pendant drop (black line). **d** Solvent-dependent surface tension measurements in water, DMF, 1-P, 1-H. **e** Schematic illustration of Ag octahedra adsorbing to the air/liquid interface; subsequent solvent evaporation lead to dual-structure supercrystals. Error bars in **c** and **d** are s.d. of at least 10 measurements

acid-stabilized PbSe nanocubes[44], and oleylamine-stabilized ZnS hexagonal bipyramids/bifrustums[45]. These prior studies show that it is thermodynamically favorable for nanoparticles to remain solvated within the liquid phase during self-assembly, because exposing the nanoparticles to the air/liquid interface require significant additional energy input. Consequently, the main particle orientation observed in our topmost layers enables an optimal amount of PVP as well as the entire particle to remain solvated within the liquid phase at the air/liquid interface during self-assembly.

The second micro-environment in our self-assembly experiments lies within the fluid phase near the drying front, where the densest-packed supercrystals form. Convective solvent flow

creates a non-equilibrium distribution of nanoparticles within the drying droplet with much higher concentration of nanoparticles localized at the drying front (Supplementary Fig. 15). Such high nanoparticle concentration in turn drives the nanoparticles to adopt hard particle behaviors, where nanoparticle morphology alone determines the type of bulk crystal structure observed at the end of the experiment[46,47]. This behavior is further supported by the relative solvent insensitivity of supercrystal formation (Supplementary Figs. 16–19, Supplementary Table 3). High local nanoparticle concentration in the experiment is akin to high pressure states commonly discussed in the simulations, with the spontaneous formation of the densest-pack crystal when the nanoparticles are suitably dispersed in a good solvent. One exception we observe is the aggregation of PVP-stabilized octahedra in toluene (Supplementary Fig. 20). Toluene is a poor solvent for PVP and aggregates can be observed immediately upon the introduction of toluene to disperse the octahedra. Unlike the open-structured topmost layers, the supercrystals observed in the bulk maximizes their face-to-face contact, suggesting that directional entropic forces may also play a part in such systems[48]. Notably, a single Ag octahedron in the Minkowski lattice is in contact with 14 other nearest neighbors via three types of face-to-face alignment (Supplementary Fig. 2).

In addition, dynamic particle transport arising from convective solvent flow toward the drying front leads to a characteristic tilt of 55–60° for all densest-packed supercrystals with respect to the substrate normal (Supplementary Fig. 21). This characteristic tilt imply that drying forces play a significant role in directing their formation in our self-assembly system. We also note that formation of our dual-structure supercrystals are concentration-dependent; self-assembly experiments using 5-fold diluted nanoparticle dispersion gives rise to nanoparticle monolayers with stable face-to-face configurations rather than open structures (Supplementary Fig. 22).

We note a decreasing extent of distinction between the topmost layer and the bulk supercrystal as the building block morphology changes from octahedra to nanocubes. Nanocubes exhibit the smallest distinction, with the topmost layer appearing nearly congruous with the bulk (Supplementary Fig. 8). In addition to changes in particle shape, the particle size also decreases from octahedra to nanocubes. Consequently, nanocubes can potentially exhibit slightly different phase behaviors from octahedra in our self-assembly system, which can be impacted by changes in interfacial energy, capillarity, and particle buoyancy in the bulk. Even though we have substantiated the importance of the air/liquid interface in driving open structure formation, how dual-structure supercrystals form remains an open question. While we are inclined towards the notion of a semi-crystalline monolayer formation at the air/liquid interface based on our experimental findings (Supplementary Figs. 11–14) and on existing literature, there is also a likelihood for smaller particles such as nanocubes at the air/liquid interface to 'merge and fit' with the bulk supercrystal during drying. In the absence of more concrete evidence, we tentatively exclude the supercrystal assembled using nanocubes as a dual-structure supercrystal due to the similarity between the topmost layer and the bulk. Nevertheless, our findings show the complexity of the self-assembly of Ag polyhedra under dynamic conditions such as in the presence of convective solvent flow within a drying droplet.

### Dual-structure supercrystal with enhanced SERS capability.

With the ability to spontaneously organize nanoparticles into two distinct structures over large areas, we demonstrate that such structures boost the overall SERS efficiency of the entire supercrystal. We use octahedra to compare the SERS efficiency between

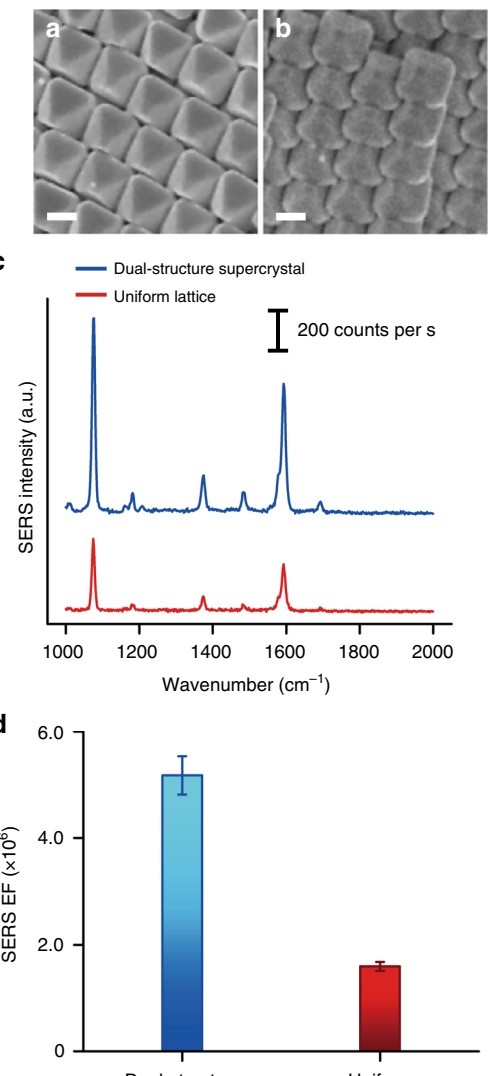

**Fig. 5** Dual-structure supercrystal generates an overall higher SERS efficiency. SEM images of the **a** dual-structure and **b** uniform bulk supercrystal of octahedra used for SERS experiments. All scale bars, 200 nm. Corresponding **c** SERS spectra and **d** EFs of 4-MBT from the respective assembled structures. Error bars in **d** are s.d. of at least 100 measurements over areas of 4 μm². 4-MBT, 4-methylbenzenethiol

the dual-structure supercrystals and the densest-packed bulk supercrystal (Fig. 5a, b). Using 4-methylbenzenthiol (4-MBT) as the probe molecule, signature vibrational modes at 1080 and 1600 cm$^{-1}$ are observed from the SERS spectra (Fig. 5c, Supplementary Table 4). The dual-structure assembly formed using octahedra generates higher signals of (962 ± 67) counts per second as compared to the (366 ± 19) counts per second of the Minkowski lattice (error bars are s.d. of at least 100 measurements). Indeed, the SERS enhancement factor (EF) for the 1080 cm$^{-1}$ mode of the octahedra dual-structure is 5.2 × 10$^6$, and is ~3.3-fold higher than the Minkowski lattice (Fig. 5d, Supplementary Note 4). Notably, this higher SERS EF arises despite the significantly lower packing density of 49% for the topmost layer of the dual-structure assembly as compared to the Minkowski lattice's 95%. Our results collectively highlight the synergistic effect of integrating two assembled structures into one supercrystal for enhanced optical sensitivity via a particle-efficient approach.

## Discussion

Our work successfully utilizes the dual self-assembly micro-environments created by a drying droplet of nanoparticle dispersion to achieve one assembly with two structures for a family of shape-controlled Ag polyhedra. These dual-structure supercrystals are in contrast with prior reports of using similar self-assembly approaches, which have predominantly resulted in either large-area monolayers of nanoparticles[2,49] and discrete supercrystals[17,25] with uniform structures. In the formation of nanoparticle monolayers, the building blocks are spherical nanoparticles (<10 nm) which form close-packed structures at the air/liquid interface. Their isotropic geometry is unable to generate structural diversity in the self-assembled structures as compared to shape-controlled nanoparticles. Discrete supercrystals with uniform structures have been achieved using surfactant-capped shape-controlled Au nanoparticles. The presence of surfactants such as CTAB can stabilize the air/liquid interface in place of the Au nanoparticles and at the same time, induce depletion attraction forces within the drying droplet through micelle formation to create the eventual discrete supercrystals. On the other hand, besides the PVP chains on the Ag polyhedra, our assembly system does not contain significant amounts of free PVP polymers because the Ag polyhedra dispersion are subject to centrifugation and redispersion prior to self-assembly experiments. Furthermore, the Ag polyhedra have edge lengths >100 nm, and can significantly decrease the interfacial energy upon adsorption. In conjunction with the structural diversity imparted by the building block morphology, we form various supercrystals with dual structures in this work.

Finally, dual-structure supercrystals can be assembled over a wide range of experimental conditions, thus establishing a versatile approach to fabricate complex crystal structures for various applications. Our work therefore opens up the possibility of new crystal design opportunities through the use of dynamic self-assembly with shape-controlled nanoparticles. Besides demonstrating the ability to achieve one assembly with two structures, we highlight its superior SERS efficiency over a uniform supercrystal. Synergistic nanoscale light-matter interactions in our crystal with dual structures enable higher SERS efficiency over a uniform supercrystal.

## Methods

**Chemicals**. Silver nitrate (≥99%), 1,5-pentanediol (PD, ≥97.0%), poly(vinyl pyrrolidone) (PVP, average $M_w$ = 55,000), 1-propanol (≥99.5%), and 1-hexanol (≥98%) were purchased from Sigma-Aldrich; copper (II) chloride (≥98%) was purchased from Alfa Aesar. All chemicals were used without further purification. Milli-Q water (>18.0 MΩ.cm) was purified with a Sartorius Arium® 611 UV ultrapure water system.

**Synthesis and purification of shape-controlled Ag polyhedra**. The synthesis of shape-controlled Ag polyhedra was carried out via the polyol reduction route[40], starting first with the synthesis of Ag nanocubes. In a typical nanocube synthesis, 10 mL of $CuCl_2$ (8 mg mL$^{-1}$), PVP (20 mg mL$^{-1}$) and AgNO$_3$ (20 mg mL$^{-1}$) were separately dissolved in PD. 35 μL of $CuCl_2$ solution was added to the AgNO$_3$ solution. 20 mL of PD was then heated to 190 °C for 10 min. 250 μL of PVP precursor was added to flask dropwise every 30 s while 500 μL AgNO$_3$ precursor was injected into the flask every minute in one go. The reaction was allowed for ~20 min. After the synthesis of nanocubes was completed, the injection was continued using more concentrated precursor solutions. 10 mL of $CuCl_2$ (8 mg mL$^{-1}$), 30 mL of PVP (20 mg mL$^{-1}$), and 30 mL of AgNO$_3$ (40 mg mL$^{-1}$) were separately prepared in PD for the synthesis of truncated nanocubes, cuboctahedra, truncated octahedra, and octahedra. After fully dissolving the AgNO$_3$ precursors, 120 μL $CuCl_2$ solution was added into the Ag precursor solution. The nanocube solution in PD was heated to 190 °C and the PVP and Ag precursors were alternately injected into the flask following the same procedure as described earlier. The reaction was allowed to proceed for 1–2 h, depending on the morphology required. For purification, the various Ag polyhedra solutions were separately re-dispersed in 20 mL ethanol after removing the PD via multiple centrifugation rounds, and diluted to ~200 mL using an aqueous PVP solution (0.2 g L$^{-1}$). This solution was then vacuum-filtered multiple times using PVDF filter membranes (Durapore®) with

pore sizes ranging from 5000, 650, 450, and 220 nm to remove impurities before finally dispersing in ethanol.

**Self-assembly of shape-controlled Ag polyhedra**. The solvent evaporation-driven self-assembly method was used for the self-assembly of various Ag polyhedra. 10 μL of 2.5 mg μL$^{-1}$ Ag polyhedra dispersion was drop-casted onto a Si substrate, and the droplet was subsequently allowed to dry at ~22 °C with a relative humidity of 50%. The time taken for the droplet to dry depended on the solvent used, ranging from ~12 h (for water and 1-propanol) to around 3 days (for N,N-dimethylformamide (DMF) and 1-hexanol).

**Characterization**. The samples after self-assembly were directly characterized using scanning electron microscopy (SEM) (JEOL-JSM-7600F). Order analyses of the various assembled arrays were characterized using the freeware ImageJ®. The radial distribution function profiles were analyzed by converting the SEM images into a binary format, followed by the automatic identification of the centers of the Ag polyhedra. Orientational analyses were performed using the deuterium order parameter[50]. One-dimensional edge vectors of Ag polyhedra were extracted and compared against the main crystal direction to determine the relative orientational order. The pendant drop method was used to measure the surface tension of the nanoparticle dispersion in various solvents as a function of volume and droplet age using an Attension Theta Lite tensiometer. The image of an ~5-μL droplet hung on the dosing needle was captured by the USB 2.0 digital camera and was subsequently analyzed by the OneAttension software. Young-Laplace model was used for droplet profile fitting during the analysis of the droplet shape to compute the surface tension for the liquid droplet. Average surface tension and standard deviation were calculated from five individual measurements at normal temperature and pressure with relative humidity of 70%. Similar measurements were conducted for the pure solvent droplets.

**SERS experiments**. SERS measurements were performed using the $x$–$y$ imaging mode of the Laser Raman Microscope RAMANtouch system with an excitation wavelength of 532 nm (power = 6 μW). A ×100 (N.A. 0.9) objective lens with 0.5 s accumulation time was used for data collection between 200 and 1800 cm$^{-1}$. $x$–$y$ SERS imaging measurements were collected from at least 100 spots with the area of each spot spanning 4 μm$^2$. The probe molecule used was 4-methylbenzenethiol and was ligand-exchanged onto the assembled structures.

**Data availability**. The authors confirm that all data supporting the findings of this study are available within the paper and its supplementary information files.

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

## Acknowledgements

X.Y.L. thanks the funding support from Singapore Ministry of Education, Tier 1 (RG21/16) and Tier 2 (MOE2016-T2-1-043), and Nanyang Technological University. H.K.L. and C.L.L. thank the A*STAR Graduate Scholarship support from A*STAR, Singapore. S.L. thanks the funding support from Academic Research Fund Tier 1 (RG107/15).

## Author contributions

Y.H.L., X.Y.L. designed research; Y.H.L., C.L.L., Y.Y.J. performed research; Y.H.L., H.K.L., W.X.S., S.Z.L., X.Y.L. analyzed the data; Y.H.L., X.Y.L. discussed and wrote the manuscript. All authors read and commented on the manuscript.

## Additional information

**Competing interests:** The authors declare no competing interests.

