## [Peer Review File · Nature Communications]

Reviewers' comments:

Reviewer #1 (Remarks to the Author):

The authors describe the use of a convective assembly approach to generate Ag nanoparticle superlattices which have a surface structure that is different from the bulk structure. These surface mesophases are somewhat analogous to a surface reconstruction in atomic crystals although the mechanism for formation in the two systems are quite different. In the present work, the authors claim that the bulk crystal forms due to hard-particle type packing behavior at the solvent evaporation front while a separate phase forms at the air-liquid interface that is more liquid crystalline in nature. Although it's not entirely clear, I believe the claim is that this loosely packed interfacial phase then rests atop the bulk crystal as the solvent evaporates at the end of the assembly process. These two-phase crystals are shown for several different nanoparticle shapes and exhibit higher SERS enhancement factors than more conventional densely-packed nanoparticle supercrystal surfaces. Overall I thought this work was high quality and parts of it were very interesting. However, I am somewhat skeptical of the novelty of most of the paper and some of the claims seem to me to be overreaching (see specific points below). If I have misunderstood the intention of the authors then I am happy to read a response but as it currently stands I do not see this as being acceptable for publication.

- The terms "translational order" and "orientational order" when used to describe the supercrystals composed of octahedra are very confusing. The authors claim that the topmost layer of octahedra (which is a different structure from the bulk) has long-range orientational order but lacks translational order. I am having trouble understanding this claim. Figure 1d and Figure 1f appear to show both translational order and orientational order. Of course the translational order in the case of the topmost layer is shorter range than the bulk (as evidenced by the radial distribution function) but it is not absent. Also, it seems to me that the origin of both the translational and orientational order of the topmost layer originates from the underlying bulk crystal. So if the translational order is diminished then the orientational order should also be diminished. This point becomes particularly confusing when the authors describe the topmost layer as being "a liquid crystal of octahedra, which lacks translational order" or "a liquid crystalline phase of the Minkowski lattice". I'm not completely sure what these phrases mean since the layer does have (diminished) translational order. I wonder if the authors could clarify these points and perhaps be a little more precise in their use of language. It might be nice to see some kind of correlation function for both the translational and the orientational order of the topmost and bulk structures. Only by comparing both of these correlation functions can the claim regarding orientational vs translational order be properly evaluated.

- I am quite convinced of the novelty of the first set of data presented on octahedra (Figure 1). It is clear that the surface layer of particles is adopting some different sort of phase compared to the underlying Minkowski lattice. However, the data for the remaining nanoparticle shapes does not appear to follow the same pattern. As far as I can tell, the cubes, truncated cubes, cuboctahedra, and truncated octahedra all appear to form a single bulk crystal that is misoriented with respect to the underlying substrate, causing an unusual surface termination that is viewed in the top-down SEM images. This point was actually quite difficult to ascertain given how the paper is written but the key data appear to be in Figure S17 which shows that all the crystals have a 55-60 degree tilt with respect to the substrate. For example, in the case of the cubes, rather than forming a large single crystal with all {100} faces being flat against the substrate as one might expect, the crystal instead is oriented with something like the [111] direction being perpendicular to the substrate. This results in a {111} surface termination of the supercrystal composed of cubes, generating the corrugated pattern shown in Figure 2a panel (iii). Consequently, rather than the surface monolayer being an entirely different structure (as in the case of octahedra), it is merely a defective or disordered surface of the underlying

bulk crystal. The authors have consistently used language to describe this case that I believe is somewhat misleading. It is described as a different crystal structure, having dual structures, having non-close-packed tiling, having edge-to-edge contact modes, and as being distinct from the thermodynamic face-to-face arrangement of cubes. The authors use the language of “plastic crystal” to describe all of these surface structures since they having varying degrees of orientational disorder but to use this distinction to claim that the surfaces are completely separate phases is claiming too much in my opinion. It seems to me that the surfaces are simply defective because of the corrugated surface structure imposed by the tilted crystal orientation. Some particles adopt the incorrect orientation and it follows that the shape of the particle would play some role in determining the degree of orientational order (Figure 2e). It is also straightforward that these misoriented and defective surfaces would have a lower packing density from the bulk. This is a serious problem for the paper, since one of the major claims is that their method is general and produces assemblies with two different crystal structures. But if one crystal structure is the bulk and the other is a disordered surface then I believe this claim is overreaching and the novelty of the manuscript is diminished. If I have misinterpreted these data then I am happy to revise this criticism.

- I am somewhat persuaded by the evidence regarding the formation of an interfacial phase of Ag nanoparticles at the air-liquid interface but I think this could be much more convincing. The authors have shown (a) that the particles separate to the air-liquid interface via surface tension measurements and (b) that in some cases the surface structure can be different from the bulk crystal. But what is missing is the evidence that the order that is adopted in the interfacial phase is translated to the topmost layer of the crystal. It seems possible that a completely disordered but liquid-like interfacial phase could produce an ordered topmost layer when slowly allowed to interact with the corrugated surface of a bulk supercrystal. Yet the authors describe the particles in the air-liquid mesophase as having precise orientations, e.g. “feature edge-to-edge contact”. While this might be true, I see no evidence for order at the air-liquid boundary and I believe it is making too many assumptions to simply look at the topmost layer in the SEM images and conclude that the interfacial mesophase has the same structure. Perhaps they can do some form of cryo TEM to freeze this structure in place or small angle X-ray scattering to determine the order at the interface.

- I’m not sure the SERS data is really necessary for this paper but if the authors want to include it I would like to see some more thorough explanation for why they observe a higher enhancement factor for the plastic crystal surface compared to a more typical bulk crystal. Yes they will have higher field enhancement at tip-to-tip junctions but fewer molecules will be able to experience this larger field. In the face-to-face junctions of the simple cubic lattice the field intensity may be lower but more molecules will experience the enhancement volume. Also, how does their measurement account for differences in the number of molecules being probed in the two experiments? I would somehow need to be convinced that the same number of molecules is being probed in both supercrystals in order to conclude that the reason for the higher signal in one is because of higher field enhancement.

Overall I think this is a very nice paper and has some fantastic data, particularly that which is summarized in Figure 1. For the octahedra, it is clear that the system adopts different bulk and surface structures. However, I’m somewhat skeptical that this is also true for the other Ag nanoparticle shapes. I’m afraid that many of the crystals were simply made in a way that they have a defective surface. I am also somewhat concerned about the strength of the data in support of the formation mechanism involving mesophases at the air-liquid interface. Because I believe the manuscript is potentially interesting I would like to give the authors a chance to address these concerns. However, if my interpretation of their data is correct then I do not see the novelty being appropriate for Nature Communications.

Reviewer #2 (Remarks to the Author):

Creating Two Self-Assembly Micro-Environments to Achieve Supercrystals with Dual Structures Using Polyhedral Nanoparticles.

The authors show nice self-assembly of Ag NP building blocks in different shapes: nanocubes, truncated cubes, cuboctahedra, truncated octahedra and octahedra. They use two different techniques to assemble the Ag NPs: one is drop casting on solid substrate and the second is by liquid-air interface. The main work was to compare between the top layer of the assembly and the inner part of the crystals. Moreover, the authors claim that the inner assembly parts correspond to Minkowski lattices.

The paper is written well, and the SEM images are quite nice, however, at this stage, I would expect to do more analysis in order to get deep understanding on such crystals. The paper/work is not unique and the authors did not show any special properties or phenomenon of this crystals, necessary to publish in Nature family. There are several similar works that already published, for example:

1. Wang, H.; Lee, H.-W.; Deng, Y.; Lu, Z.; Hsu, P.-C.; Liu, Y.; Lin, D.; Cui, Y. Nat. Commun. 2015, 6, 7261.

- I recommend to synchronize between figure 2 and the main text (or to split figure 2). The figure is very busy and the reader loses the principle during reading.
- Correct the name in the main text (Fig. 4 instead Fig.5)

Reviewer #3 (Remarks to the Author):

The authors present a detailed experimental study showing that Ag nanopolyhedra can be assembled into dual structures, where the order of the top layer differs from the rest of the supercrystal, via simple drying of a droplet placed on a Si surface. In addition, they show that such structures can result in SERS enhancement. This results is new to me, and I expect it to be of strong interest to others in the fields and likely to motivate future work. The paper is clearly written and presented, with appropriate reference to previous literature, and sufficient methodological detail. That said, a number of claims should probably be clarified, amended, or supported with additional data.

1) The claim that the top layer of octahedra is a liquid crystal is technically wrong. It is neither liquid, nor does it entirely lack translational order (as claimed). In Fig. S3, typical crystal defects, including grain boundaries and vacancies, are visible. Several different types of packing, with distinct lattice parameters, are also visible. The long-range translational order appears to be reduced due to this polydispersity and perhaps a lattice mismatch between the top layer and the underlying Minkowski lattice. The claim that the topmost layer is "a liquid crystalline phase of the Minkowski lattice" also makes no sense.

2) Calling the dual structure a "single supercrystal" is also inaccurate. It consists of one ordered structure deposited on top of a different one.

3) The sentence "Moreover, the dominant nanoparticle orientations in the mesophases correspond to regions with large curvatures on the respective nanoparticle morphologies" needs clarification or rewriting. Looking at the results, and considering what drives colloids to fluid-fluid interfaces, it seems at least equally likely that the particles are simply oriented in such a way that they maximise the area of interface that each particle displaces, with some variation due to orientational entropy.

4) It is difficult to justify the argument that directional entropic forces are responsible for formation of the Minkowski lattice when there is relatively little face-to-face contact between octahedra in this structure. Indeed, adding a depletant results in an increase in face-to-face contact and a change to a less dense lattice, as demonstrated in Ref. 14.

5) What does the following sentence mean and why is it justified? "At the same time, the orientation of the Ag polyhedra in the tilted bulk supercrystals coincides with the main particle orientation observed among the mesophase in the topmost layer." Particles in the topmost layer adopt several different orientations. Do they all coincide with the orientation of polyhedra in the underlying supercrystal and, if so, how precisely?

6) The arguments that dual structures lead to SERS enhancement in general and that this is due to synergistic effects are not adequately supported by the data. In particular, it is unclear whether shapes other than cubes will give rise to SERS enhancement, and it is unclear whether the enhancement is simply due to the spikier top layer rather than a property of the dual structure.

Reviewers' comments:

Reviewer #1 (Remarks to the Author):

The authors describe the use of a convective assembly approach to generate Ag nanoparticle superlattices which have a surface structure that is different from the bulk structure. These surface mesophases are somewhat analogous to a surface reconstruction in atomic crystals although the mechanism for formation in the two systems are quite different. In the present work, the authors claim that the bulk crystal forms due to hard-particle type packing behavior at the solvent evaporation front while a separate phase forms at the air-liquid interface that is more liquid crystalline in nature. Although it's not entirely clear, I believe the claim is that this loosely packed interfacial phase then rests atop the bulk crystal as the solvent evaporates at the end of the assembly process. These two-phase crystals are shown for several different nanoparticle shapes and exhibit higher SERS enhancement factors than more conventional densely-packed nanoparticle supercrystal surfaces. Overall I thought this work was high quality and parts of it were very interesting. However, I am somewhat skeptical of the novelty of most of the paper and some of the claims seem to me to be overreaching (see specific points below). If I have misunderstood the intention of the authors then I am happy to read a response but as it currently stands I do not see this as being acceptable for publication.

We thank the reviewer for the positive and constructive feedback provided over here and below are our point-by-point response to the concerns raised by the reviewer.

- The terms “translational order” and “orientational order” when used to describe the supercrystals composed of octahedra are very confusing. The authors claim that the topmost layer of octahedra (which is a different structure from the bulk) has long-range orientational order but lacks translational order. I am having trouble understanding this claim. Figure 1d and Figure 1f appear to show both translational order and orientational order. Of course the translational order in the case of the topmost layer is shorter range than the bulk (as evidenced by the radial distribution function) but it is not absent. Also, it seems to me that the origin of both the translational and orientational order of the topmost layer originates from the underlying bulk

crystal. So if the translational order is diminished then the orientational order should also be diminished. This point becomes particularly confusing when the authors describe the topmost layer as being “a liquid crystal of octahedra, which lacks translational order” or “a liquid crystalline phase of the Minkowski lattice”. I’m not completely sure what these phrases mean since the layer does have (diminished) translational order. I wonder if the authors could clarify these points and perhaps be a little more precise in their use of language. It might be nice to see some kind of correlation function for both the translational and the orientational order of the topmost and bulk structures. Only by comparing both of these correlation functions can the claim regarding orientational vs translational order be properly evaluated.

We thank the reviewer for this constructive comment. We agree with the reviewer that there is a decrease in translational order for the topmost layer as compared to the bulk supercrystal in the Ag octahedra system. We will also show that this topmost layer does not exhibit reduced orientational order in comparison with the bulk supercrystal. Based on these two criteria, we therefore classify the topmost layer of Ag octahedra as a liquid crystal. We have also performed radial distribution function (RDF) and the deuterium order parameter (SCD) to systematically correlate the respective translational and orientational order of the topmost and bulk structures. Regarding the comment on the origin of the top layer, we will be discussing our hypothesis, new analyses together with new supporting information in the next two questions.

Let us first clarify the various terms used in this manuscript. Nanoparticles can undergo various pathways and exhibit diverse phase behaviors during the transition from the isotropic fluid state to the crystalline state. In the isotropic state, nanoparticles are randomly dispersed within a solvent (experimentally) or within a simulation box (theoretically). Mesophases, in the form of plastic or liquid crystals, can arise during this transition from isotropic fluid state to the crystalline state. In plastic crystals, nanoparticles lose their orientational order while maintaining translational order. On the other hand, nanoparticles in liquid crystals lose their translational order while maintaining orientational order (Figure R1-1, *Nat. Mater.* **2011**, 10, 230; *Science*, **2012**, 337, 453). It should be noted that this loss in order is relative, and can range from a slight loss in order, to diminished and reduced order, as per the reviewer’s observations.

Figure R1-1. Various possible phase behaviors of nanoparticles between isotropic and crystalline states. Reprinted with permission from *Nat. Mater.* **2011**, 10, 230. Copyright 2011 Nature Publishing Group.

In our system, to better correlate the translational and orientational order of our assembled metacrystals, we employ two software processing techniques, namely radial distribution function (RDF) and the deuterium order parameter (S_{CD}), on our SEM images. RDF analyzes the translational ordering of the metacrystals, while S_{CD} analyzes the orientational ordering.

For the Ag octahedra system, the topmost layer is a liquid crystal. The RDF profiles examining the translational ordering of the topmost layer and bulk supercrystal are shown in Fig. 1d and f in the manuscript respectively (Figure R1-2). The topmost layer has lower translational order than the bulk supercrystal, as the reviewer pointed out from his/her observations. The lower translational order is evident from our RDF analysis as the topmost layer has lesser peaks in its RDF profile than the bulk supercrystal. Peaks are also sharper and more defined in the bulk supercrystal than the topmost layer. In the bulk supercrystal, the separation distances between neighboring particles are highly defined and regular, giving rise to discrete and intense peaks in the RDF profile (*J. Chem. Phys.* **2004**, 120, 9383). In contrast, nanoparticles in the topmost layer are displaced from the ideal lattice positions, thus generating broader and less intense peaks in its RDF profile. The broader peaks arise from a distribution of separation distances from the ideal

separation. Furthermore, we note that octahedra in the topmost layer are aligned in the same direction, analogous to the smectic phases of liquid crystals observed in Au nanorods (*Angew. Chem. Int. Ed.*, **2008**, 47, 9685).

Figure R1-2. Dual crystal structure formed using Ag octahedra in water. (a, b) Cross-sectional panoramic characterization shows the formation of large-area supercrystals with long-range order. The heights of the supercrystal are nearly uniform at $\sim 10 \mu\text{m}$ and stretching more than $50 \mu\text{m}$ across. (c-h) The supercrystal can be divided into two distinct regions. (c, d) The top layer features a liquid crystal with Ag octahedra possessing long-range orientational order without translational order. (e, f) The bulk of the supercrystal is crystalline, and can be indexed to the Minkowski lattice (inset of e). Insets of d, f are the FFT images of the respective crystal structure. The Ag octahedra in the liquid crystalline phases are color-coded for ease of discussion. (g) Radial distribution function of the respective crystal.

To analyze the orientational ordering of the metacrystals, we perform additional image analyses using the deuterium order parameter (S_{CD}). S_{CD} is typically used to evaluate the orientational order of hydrocarbon tails in phospholipid bilayers (*Eur. Biophys. J.*, **2007**, 36, 919). S_{CD} is calculated using the equation $S = (3\cos^2\theta - 1)/2$. A value of $S = 1$ indicates perfect nanoparticle orientation along the crystal direction whereas $S = 0.5$ corresponds to a nanoparticle orientation perpendicular to the crystal direction. $S = 0$ indicates a random nanoparticle orientation with respect to the crystal direction. θ refers to the angle between the nanoparticle edge vector and the main crystal direction (Figure R1-3, yellow arrows and red dash lines). For our system, we compare the percentage difference in S values ($\% \Delta S_{CD}$) obtained for the topmost layer with that obtained for the bulk supercrystal to correlate the relative loss in orientational order for the topmost layer.

Figure R1-3. Orientational order analyses for Ag octahedra. SEM images labeled with the nanoparticle edge vectors for the (i) topmost layer and (ii) bulk supercrystal in yellow. Yellow arrows indicate the edge vector direction. Red dash lines indicate the main orientation of the assembled structure. (iii) Orientational order for the topmost and bulk supercrystal.

To adapt S_{CD} for our nanoparticle-based system, we manually label the nanoparticles in the SEM images to extract the cartesian coordinates (Figure R1-3, marked in yellow in panels i and ii). The coordinates are then used to derive one-dimensional edge vectors; these edge vectors represent the orientation of the nanoparticle edge relative to a pre-defined cartesian axis. In our

analysis, we choose the main crystal direction of the assembled structure as the default axis (Figure R1-3, red dashed lines panels i and ii). By comparing the edge vectors with the main crystal direction, we can obtain an angle θ to describe the nanoparticle orientation with respect to the main crystal direction.

For the octahedra system, $S = 0.9602$ and $S = 0.9565$ for the bulk supercrystal and topmost layer respectively (Figure R1-3). The orientational ordering of both the topmost layer and bulk supercrystal differs by only 0.4 %, indicating negligible loss of orientational order in the octahedra system. In addition, the standard deviation of S values for both the topmost layer and the bulk supercrystal are in the same range of ~ 10 %, indicating similar distribution of orientational orderness for both structures. Together with the reduced translational order, and the negligible loss of orientational order, we classify the topmost layer of the Ag octahedra as a liquid crystalline phase using octahedral particles as building blocks. This classification also agrees well with the reviewer's perspective.

In addition to our work here, various types of liquid and plastic crystals have been reported using micro-/nanoparticles, including nanorods (*Phys. Rev. Lett.*, **2003**, 90, 018303), nanoplates (*ACS Nano*, **2011**, 5, 8322), and various superball colloids (*Nat. Commun.*, **2017**, 8, 14352). In all these works, various types of ordering have been observed, with some possessing high translational and/or orientational order.

To refine our discussion in our manuscript, we have made the following changes:

Page 5, Paragraph 1:

'In contrast, octahedra in the topmost layer organize into a liquid crystal, with reduced translational order (Fig. 1 b-d, blue colored layer).'

'...FFT analysis of the topmost layer also demonstrates the crystallinity of this structure (insets of Fig. 1d, f), with the experimentally determined packing efficiency estimated to be 49 % (Fig. S4). However, a displacement in the lattice positions of the octahedra indicates a loss of translational order.'

‘On the other hand, long-range orientational order is evident for this top layer, with octahedra oriented along the same direction. Orientational order analysis using the deuterium order parameter indicates that there is negligible reduction of orientational order between the topmost layer and the Minkowski lattice (Table S1).’

The following definitions have also been supplemented with Table S1 (Page 5 of Supplementary Information):

‘Nanoparticles can undergo various pathways and exhibit diverse phase behaviors during the transition from the isotropic fluid state to the crystalline state. In the isotropic state, nanoparticles are randomly dispersed within a solvent (experimentally) or within a simulation box (theoretically). Mesophases, in the form of plastic or liquid crystals, can arise during this transition from isotropic fluid state to the crystalline state. In plastic crystals, nanoparticles lose their orientational order while maintaining translational order. On the other hand, nanoparticles in liquid crystals lose their translational order while maintaining orientational order. It should be noted that this loss in order is relative, and can range from a slight loss in order, to diminished and reduced order.’

To analyze the orientational ordering of the metacrystals, we perform additional image analyses using the deuterium order parameter (S_{CD}). S_{CD} is typically used to evaluate the orientational order of hydrocarbon tails in phospholipid bilayers and is calculated using the equation $S = (3\cos^2\theta - 1)/2$. A value of $S = 1$ indicates perfect nanoparticle orientation along the crystal direction whereas $S = 0.5$ corresponds to a nanoparticle orientation perpendicular to the crystal direction. $S = 0$ indicates a random nanoparticle orientation with respect to the crystal direction. θ refers to the angle between the nanoparticle edge vector and the main crystal direction.

To adapt S_{CD} for our nanoparticle-based system, we manually label the nanoparticles in the SEM images to extract the cartesian coordinates. The coordinates are then used to derive one-dimensional edge vectors; these edge vectors represent the orientation of the nanoparticle edge relative to a pre-defined cartesian axis. In our analysis, we choose the main crystal direction of

the assembled structure as the default axis. By comparing the edge vectors with the main crystal direction, we can obtain an angle θ to describe the nanoparticle orientation with respect to the main crystal direction. We compare the percentage difference in S values ($\% \Delta S_{CD}$) obtained for the topmost layer with that obtained for the bulk supercrystal to correlate the relative loss in orientational order for the topmost layer.'

- I am quite convinced of the novelty of the first set of data presented on octahedra (Figure 1). It is clear that the surface layer of particles is adopting some different sort of phase compared to the underlying Minkowski lattice. However, the data for the remaining nanoparticle shapes does not appear to follow the same pattern. As far as I can tell, the cubes, truncated cubes, cuboctahedra, and truncated octahedra all appear to form a single bulk crystal that is misoriented with respect to the underlying substrate, causing an unusual surface termination that is viewed in the top-down SEM images. This point was actually quite difficult to ascertain given how the paper is written but the key data appear to be in Figure S17 which shows that all the crystals have a 55-60 degree tilt with respect to the substrate. For example, in the case of the cubes, rather than forming a large single crystal with all $\{100\}$ faces being flat against the substrate as one might expect, the crystal instead is oriented with something like the $[111]$ direction being perpendicular to the substrate. This results in a $\{111\}$ surface termination of the supercrystal composed of cubes, generating the corrugated pattern shown in Figure 2a panel (iii). Consequently, rather than the surface monolayer being an entirely different structure (as in the case of octahedra), it is merely a defective or disordered surface of the underlying bulk crystal. The authors have consistently used language to describe this case that I believe is somewhat misleading. It is described as a different crystal structure, having dual structures, having non-close-packed tiling, having edge-to-edge contact modes, and as being distinct from the thermodynamic face-to-face arrangement of cubes. The authors use the language of "plastic crystal" to describe all of these surface structures since they having varying degrees of orientational disorder but to use this distinction to claim that the surfaces are completely separate phases is claiming too much in my opinion. It seems to me that the surfaces are simply defective because of the corrugated surface structure imposed by the tilted crystal orientation. Some particles adopt the incorrect orientation and it follows that the shape of the particle would play some role in determining the degree of orientational order (Figure 2e). It is also straightforward that these misoriented and defective surfaces would have a

lower packing density from the bulk. This is a serious problem for the paper, since one of the major claims is that their method is general and produces assemblies with two different crystal structures. But if one crystal structure is the bulk and the other is a disordered surface then I believe this claim is overreaching and the novelty of the manuscript is diminished. If I have misinterpreted these data then I am happy to revise this criticism.

First, we emphasize that the topmost layers of nanocubes, truncated nanocubes, cuboctahedra, and truncated octahedra are plastic crystals. Their topmost layers exhibit crystal structures distinct from the bulk, possessing translational order. In addition, they present various extents of orientational disorder which we will discuss in greater detail in the following paragraphs. Based on our definition of the mesophase in question 1 above, these topmost layers can be categorized as plastic crystals with reduced orientational order. In the second section, we will deliberate that these topmost layers are not a mis-orientation/ defect of the bulk. Based on the supports from correlation function analyses, SEM observations and analyses, control experiments under various conditions, we propose that such topmost layers are the results of the competition between nanoparticle interfacial adsorption behavior and the subsequent combination with the bulk supercrystal during solvent drying.

Our stand on “nanocubes, truncated nanocubes, cuboctahedra, and truncated octahedra are plastic crystals” is based on our results in Figure 2 of the manuscript. For ease of discussion, we will first use nanocube as an example, and extend our discussion to the other shaped particles in a latter section. Nanocubes adopt hexagonal structures on the topmost layer, whereas their bulk supercrystals are spacing-filling square lattices (Figure R1-4a). This distinction in crystal structures are evident from the visual examination of the SEM images, and from the fast-Fourier transform analyses where the topmost layer and bulk supercrystal generate different diffraction spots. The hexagonal structures of the nanocubes in the topmost layer generate six diffraction spots in its FFT, which is clearly distinct from the four diffraction spots in the FFTs of its corresponding bulk supercrystal (Figure R1-4a, panels ii vs insets of panels iv).

In addition, we perform RDF analyses for nanocubes to evaluate the translational ordering (Figure R1-5a). Long-range translational order is evident from both the topmost layers and the

bulk supercrystals. Multiple peaks are observed from the RDF profiles for both the topmost layers and the bulk supercrystals. However, there are no overlapping peaks in topmost layer and bulk, and the number of peaks present in the corresponding RDF profiles is different. This is a clear indication that the lattice structures of the topmost structures and the bulk supercrystals are distinct from each other.

Upon affirming the crystallinity of the topmost layer, we then evaluate the loss of orientational order for the nanocube plastic crystal using the deuterium order parameter (Table R1-1). As the reviewer noted, the orientational order is significantly lower for the topmost layer as compared to the corresponding bulk supercrystal, clearly indicating the plastic crystal nature of this topmost layer. The percentage difference in S_{CD} between the bulk supercrystal and the topmost layer differs by at least 18 % and this percentage certainly contrasts with that observed for octahedral nanoparticles. The lower orientational order of the topmost layer for the nanocubes thus supports our conclusion that the nanocubes on the topmost layer are indeed plastic crystals.

Figure R1-4. Dual crystal structures of Ag polyhedra. SEM characterization of (a) nanocubes, (b) truncated nanocubes, (c) cuboctahedra and (d) truncated octahedra. (i) Top-view, (ii) FFTs and (iii) close-up images of the various assembled structures. (iv) Cross-sectional characterization of the bulk supercrystals. Insets are FFTs of the crystal structures. (e) Relative population of the various orientations adopted by the mesophases of Ag polyhedra. (f) Packing densities of the various mesophases and their supercrystals.

Figure R1-5. RDF profiles of (a) nanocubes, (b) truncated nanocubes, (c) cuboctahedra and (d) truncated octahedra.

Table R1-1. Orientational order analysis of the various assembled structures using the respective building blocks.

	Truncated		Truncated		
	Cubes	Cubes	Cuboctahedra	Octahedra	Octahedra
$S_{CD} (Bulk)$	0.9806	0.8937	0.9820	0.9027	0.9602
$S_{CD} (Mesophase)$	0.8042	0.8003	0.7446	0.7948	0.9565
$\% \Delta S_{CD}$	18	11	24	12	0.4

Similarly, the topmost layer for truncated octahedra, cuboctahedra, and truncated octahedra are also plastic crystals with distinct structures from their corresponding bulk supercrystals. Truncated nanocubes adopt hexagonal structures on the topmost layer, whereas their bulk

supercrystals are spacing-filling square lattices (Figure R1-4b). Cuboctahedra in the topmost layer are organized into square structures, while their bulk supercrystals assemble into body-centered tetragonal lattices (Figure R1-4c). Truncated octahedra also form hexagonal structures on the topmost layer, and this structure is again distinct from their body-centered tetragonal lattice in the bulk (Figure R1-4d). All these crystal structure differences are evident from the visual examination of the SEM images, and from the fast-Fourier transform analyses where the topmost layer and bulk supercrystal generate different diffraction spots. The hexagonal structures of the truncated nanocubes, and truncated octahedra in the topmost layer generate six diffraction spots in their respective FFTs, which are clearly distinct from the four diffraction spots in the respective FFTs of their corresponding bulk supercrystals (Figure R1-4, panels ii vs insets of panels iv).

RDF profiles also indicate long-range translational order for both the topmost layers and the bulk supercrystals for truncated nanocubes, cuboctahedra, and truncated octahedra (Figure R1-5b-d). Multiple peaks are observed from the RDF profiles for both the topmost layers and the bulk supercrystals. There are no overlapping peaks for these nanoparticle morphologies and the number of peaks present in the corresponding RDF profiles is different. This is because the lattice structures of the topmost structures and the bulk supercrystals are distinct from each other.

The loss of orientational order for these plastic crystals are shown in Table R1. As with nanocubes, the orientational order is significantly lower for the topmost layer as compared to the corresponding bulk supercrystal for these four particle morphologies. The percentage difference in S_{CD} between the bulk supercrystal and the topmost layer differs up to 24 %. Collectively, the lower orientational order of the topmost layer for truncated nanocubes, cuboctahedra, and truncated octahedra thus supports our conclusion that the mesophases of this group of nanoparticles are indeed plastic crystals with distinct crystal structures from their bulk supercrystals. The mesophases observed in our work are by no means a mis-orientation or defects arising from the bulk.

With regards on to the reviewer's question on whether "the top surface is simply defect because of the corrugated surface structure imposed by the tilted crystal orientation of the bulk" and

whether “the origin of both the translational and orientational order of the topmost layer originates from the underlying bulk crystal” (from question 1), we once again emphasize that they are two distinct crystals based on our analytical evaluation above.

We hypothesize that the competition between nanoparticle interfacial adsorption behavior and the subsequent combination with the bulk supercrystal during solvent drying contributes to the extent of disruption observed between the topmost layer and the bulk supercrystal, and hence the clarity of the dual-structure supercrystal. This hypothesis arises from a qualitative trend observed in Figure R1-6: the disruption between the topmost layers and the bulk layers becomes increasingly evident as the particle morphology transits from nanocubes to octahedra. Our hypothesis is based on several main considerations that we will elaborate in detail in subsequent paragraphs. First, control experiments under various conditions, combined SEM observation and existing literature findings point to the fact that nanoparticles at the interface do indeed form a semi-crystalline structure. Secondly, the interfacial mobility of nanoparticles as well as the van der Waals attraction between nanoparticles at the interface is dependent on particle sizes. The combination of these factors thus give rise to orientational disorder among the nanocube plastic crystal and translational disorder among the octahedra liquid crystal as the topmost layer merges with the bulk supercrystal during drying.

Figure R1-6. Close-up cross-sectional SEM images of (a) nanocubes, (b) truncated nanocubes, (c) cuboctahedra, (d) truncated octahedra, and (e) octahedra indicating the disruption between the topmost layer of nanoparticles (highlighted in blue) and the bulk supercrystal.

The first support for our hypothesis stems from our current experimental observations combined with literature findings that various Ag polyhedra form a semi-crystalline monolayer at the air/liquid interface, where it is thermodynamically favorable for the polyhedra to minimize exposure to air during self-assembly. For nanocubes, [111] orientation is the dominant configuration observed (*Nano Lett.*, **2012**, 12, 4791). Nanoparticles spontaneously adsorb to the air/liquid interface to minimize the interfacial energy. We have also experimentally demonstrated the decreases in surface tension upon nanoparticle adsorption to the air/liquid interface, which supports the fact that nanoparticles do adsorb to the air/liquid interface in our system to minimize the interfacial energy. At the same time, it is reasonable to assume that the various Ag polyhedra

in our work remain fully solvated at the air/liquid interface during self-assembly. Simulations have shown that nanoparticles at the air/liquid interface remain completely solvated during self-assembly, and that fully exposing a ~ 30 nm nanoparticle to air requires a higher free energy input of $\sim 10^4 k_B T$ (*Nano Lett.*, **2014**, 14, 1032). As such, the associated increase in free energy for our nanoparticles to be exposed to air will be even higher, since their edge lengths are upwards of 100 nm. Furthermore, for nanocubes system, a prior study of PbSe nanocubes using small angle X-ray scattering shows that the nanocubes do indeed adopt a [111] termination at the air/liquid interface during self-assembly to minimize exposure to air (*Nano Lett.*, **2012**, 12, 4791). Based on the findings of these works, together with our SEM observations of the nanocubes, it is reasonable to assume that in our system, a semi-crystalline monolayer of nanocubes forms at the air/liquid interface during self-assembly, adopting a dominant [111] orientation to enable the nanocubes to remain maximally solvated within the solvent. This semi-crystalline monolayer of [111]-oriented nanocubes subsequently dries onto the bulk surface, giving rise to an eventual corrugated surface with slight mis-orientation. The slight angular misalignment is $\sim 5^\circ$ between the topmost layer of the nanocube plastic crystal and the bulk supercrystal (α in Figure R1-6a). This misalignment implies that the nanocubes in the topmost layer did not just ‘fit’ into the corrugated surface of the bulk supercrystal.

In addition, several control self-assembly experiments based on solvent, temperature, and various tilting angle effects suggest a high likelihood that semi-crystalline structures indeed form at the air/liquid interface prior to complete evaporation of the solvent. Using octahedra system as the model, we observe persistent formation of the liquid crystal over a wide range of experimental conditions. Liquid crystal formation occurs in a wide range of solvent polarities as long as the particles can be stably dispersed in the solvents (Figure R1-7a-c), including water (most polar, dielectric constant ~ 79.7) and 1-hexanol (least polar, dielectric constant ~ 13.3). Adjusting the geometrical setup of the self-assembly (by tilting or inverting the Si substrate after solvent droplet deposition) also results in the continued observation of liquid crystals (Figure R1-7d, e). Liquid crystal formation is also independent of self-assembly temperature and evaporation rate, and is observed to form at elevated temperatures of up to $\sim 60^\circ \text{C}$ (Figure R1-7f-h). At this temperature, complete solvent evaporation occurs in less than an hour (~ 15 min), and this short duration implies that interaction between the topmost layer and the underlying bulk supercrystal

is insufficient to allow interaction with the underlying bulk supercrystal. On the other hand, prolonging solvent evaporation to ~3 days (either by saturating the self-assembly environment with solvent or using high boiling point solvents such as N,N-dimethylformamide) also does not eliminate liquid crystal formation. In contrast, when the effect of the interface is eliminated by allowing the octahedra to sediment during self-assembly, the formation of pure Minkowski lattice is observed (*Nat. Mater.* **2012**, 11, 131). Our combined experiments here strongly imply that liquid crystals will always be observed as long as there is an air/liquid interface.

Figure R1-7. Formation of liquid crystal in (a) DMF, (b) 1-propanol, and (c) 1-hexanol. Liquid crystal is observed when the (d) droplet is dried upside down, (e) droplet is tilted at an angle, (f) droplet is dried in an environment saturated with water vapor and at elevated temperatures of (g)

40°C and (h) 60°C. In the water vapor-saturated environment, nanoparticle dispersion took ~ 3 days to complete drying. At elevated temperatures, drying is complete within an hour.

Secondly, the smaller dimensions of the nanocubes (edge length ~110 nm) in comparison to octahedra (edge length ~350 nm) imply that the nanocubes are easily subject to orientational changes during solvent drying which can cause microscale fluctuations to the solvent surface as well as during the subsequent combination with the bulk supercrystal. Such orientational changes potentially contribute to the formation of plastic crystal for the nanocubes. This is because the associated change in interfacial energy upon nanoparticle adsorption is an order of magnitude smaller for nanocubes than octahedra. To estimate the change in interfacial energy upon nanoparticle adsorption, we use the equation $\Delta E = (\gamma_0 - \gamma)\pi R^2/\eta$ (*Langmuir*, **2010**, 26, 12518). The change in interfacial energy (ΔE) is dependent on the nanoparticle radius (R) and its packing fraction (η); γ_0 and γ refer to the initial and final interfacial tension measured. Assuming all other parameters are approximately constant, a decrease in ~ 3.2-fold edge length leads to ~ 10-fold decrease in the change in interfacial energy for the nanocubes as compared to octahedra. Consequently, nanocubes at the interface are relatively more mobile than octahedra, capable of rotating and can even undergo adsorption/desorption during the solvent drying process. This mobility can also contribute to the loss in orientational order among the semi-crystalline monolayer, especially during the subsequent combination with the bulk supercrystal during solvent drying. For octahedra, the larger change in interfacial energy lowers the interfacial mobility and reduces the potential for a loss in orientational order.

Thirdly, the van der Waals attraction among neighboring particles in the topmost layer also impacts its merging with the bulk supercrystal. Van der Waals attraction between neighboring particles scale proportionally with particle size (*Small*, **2009**, 5, 1600). Neighboring nanocubes with their smaller size implies a weaker van der Waals attraction as compared to the much larger octahedra. Due to the lower additional energy required for nanocubes to rotate at the interface, as well as weaker van der Waals attraction between neighboring nanocubes, the disruption between the topmost layer of nanocubes and the bulk supercrystal is the smallest. This gives rise to a plastic crystal observed for nanocubes. On the other hand, significantly stronger van der Waals attraction arises between the much larger octahedra, thus preserving the orientation of the

monolayer and hence giving rise to the liquid crystal observed. As the solvent dries, the merging of the topmost layer with the bulk will to a certain extent disrupt the topmost layer.

Summing up our argument here, the combination of interfacial adsorption in relation to particle mobility, as well as van der Waals forces between neighboring particles gives rise to our dual-structure supercrystals as the topmost layer combines with the bulk supercrystal during solvent drying in this work. The interplay of these forces gives rise to plastic crystals for nanocubes (smaller change in interfacial adsorption energy, higher mobility, weaker van der Waals) and liquid crystals for octahedra (larger change in interfacial adsorption energy, lower mobility, stronger van der Waals).

We have added the following sentence to our manuscript:

Page 5, Paragraph 2:

‘Long-range translational order of the structures is evident from the SEM, FFT, and RDF profiles for all the plastic crystals observed (Fig. 2a-d, panel ii, Fig. S6).’

‘The reduction in orientational order among these topmost layers is further affirmed through orientational order analyses performed using the deuterium order parameter (Table S1), thus indicating that the topmost layers are plastic crystals.’

Page 6, Paragraph 2:

‘The differences observed from the FFT patterns, RDF profiles, as well as close-up SEM images for the plastic crystals and the bulk supercrystals further show that they are two distinct structures (Fig. S8).’

Page 7, Paragraph 1:

‘On the other hand, previous self-assembly experiments based on the sedimentation of Ag octahedra in the absence of an air/liquid interface did not give rise to liquid crystalline phase formation, further affirming the role of the interface in driving liquid crystal formation.’

In addition, Figure 2 has been updated to include the FFT for the bulk supercrystals, Table R1-1 has been included as Table S1 (Page 5 of Supplementary Information), Figure R1-5 has been included as Figure S6 (Page 7 of Supplementary Information), and Figure R1-6 has been included as Figure S8 (Page 9 of Supplementary Information).

- I am somewhat persuaded by the evidence regarding the formation of an interfacial phase of Ag nanoparticles at the air-liquid interface but I think this could be much more convincing. The authors have shown (a) that the particles separate to the air-liquid interface via surface tension measurements and (b) that in some cases the surface structure can be different from the bulk crystal. But what is missing is the evidence that the order that is adopted in the interfacial phase is translated to the topmost layer of the crystal. It seems possible that a completely disordered but liquid-like interfacial phase could produce an ordered topmost layer when slowly allowed to interact with the corrugated surface of a bulk supercrystal. Yet the authors describe the particles in the air-liquid mesophase as having precise orientations, e.g. “feature edge-to-edge contact”. While this might be true, I see no evidence for order at the air-liquid boundary and I believe it is making too many assumptions to simply look at the topmost layer in the SEM images and conclude that the interfacial mesophase has the same structure. Perhaps they can do some form of cryo TEM to freeze this structure in place or small angle X-ray scattering to determine the order at the interface.

We apologize for the over-claiming in the section where we discussed the ‘particle-efficient approach to stabilize the air/liquid interface...with Ag polyhedra featuring edge-to-edge contact mode’ in the manuscript. We clarify that this statement was made based on the final assembled structures that we observe from SEM characterization, not at the air-liquid interface. We have removed this statement in our revised manuscript.

Here we address the reviewer's concern on "It seems possible that a completely disordered but liquid-like interfacial phase could produce an ordered topmost layer when slowly allowed to interact with the corrugated surface of a bulk supercrystal".

We discuss a high likelihood that semi-crystalline structures indeed form at the air/liquid interface prior to complete evaporation of the solvent using several control self-assembly experiments. Using octahedra system as the model, we observe persistent formation of the liquid crystal over a wide range of experimental conditions. Liquid crystal formation occurs in a wide range of solvent polarities as long as the particles can be stably dispersed in the solvents (Figure R1-7a-c), including water (most polar, dielectric constant ~ 79.7) and 1-hexanol (least polar, dielectric constant ~ 13.3). Adjusting the geometrical setup of the self-assembly (by tilting or inverting the Si substrate after solvent droplet deposition) also results in the continued observation of liquid crystals (Figure R1-7d, e). Liquid crystal formation is also independent of self-assembly temperature and evaporation rate, and are observed to form at elevated temperatures of up to $\sim 60^{\circ}\text{C}$ (Figure R1-7f-h). At this temperature, complete solvent evaporation occurs in less than an hour. On the other hand, prolonging solvent evaporation to ~ 3 days (either by saturating the self-assembly environment with solvent or using high boiling point solvents such as N,N-dimethylformamide) also does not eliminate liquid crystal formation. In contrast, when the effect of the interface is eliminated by allowing the octahedra to sediment during self-assembly can the formation of pure Minkowski lattice be observed (*Nat. Mater.* **2012**, 11, 131). Our combined experiments here strong imply that liquid crystals will always be observed as long as there is an air/liquid interface, and that evaporation time plays a secondary role in its formation.

We further highlight that our comprehensive analyses performed in earlier discussion demonstrates that nanocubes, truncated nanocubes, cuboctahedra, and truncated octahedra form plastic crystals whereas octahedra form liquid crystals in the topmost layer after drying. For the plastic crystals, their crystal structures are distinct from the bulk, and as such, cannot be formed purely as a result of a corrugated surface. For the octahedra liquid crystal, the loss of translational order possibly arises due to the slight disruption when the topmost layer assembles onto the bulk. Hence, the crystallinity of the interfacial phase is likely to be translated to the

topmost layer, albeit with reduced orientational order as they assemble, and/or decrease in translational order upon contacting the corrugated surface.

To this end, we are aware that current experimental observations alone are insufficient to clearly elucidate the role of each micro-environment, and how one micro-environment affects the other during nanoparticle self-assembly. We have indeed carefully considered the reviewer's suggestion of using cryo-TEM or small angle X-ray scattering to determine order at the interface, but we regret that both methods are either not feasible or unavailable for our system. The reasons are stated below. First, TEM imaging requires the electron beam to be transmitted through the samples. Our nanoparticles have edge lengths > 100 nm, which are thick enough to prevent electron beam transmission to the detector. As such, cryo-TEM will not be able to provide further information on the structural configuration at the interface. We have also attempted to overcome such limitations by using liquid-based scanning electron microscopy. However, the experimental setup involves depositing a liquid droplet within a specialized cell and this removes the interface at which the nanoparticles adsorb. At the same time, solvent presence (~ 4 μ L) do not allow the electron beam to penetrate the liquid cell to probe the process of nanoparticle self-assembly in our system. Second, small angle X-ray scattering typically requires the use of a synchrotron X-ray source, and we currently have no direct access to it. Nevertheless, a prior study of PbSe nanocubes using small angle X-ray scattering shows that the nanocubes do indeed adopt a [111] termination at the air/liquid interface during self-assembly (*Nano Lett.*, **2012**, 12, 4791). Consequently, we are also unable to unambiguously determine the actual orientation of the nanoparticles at the air/liquid interface during the self-assembly process.

Nevertheless, the interface (air/liquid, liquid/liquid) has long been established as a versatile platform to assemble and organize nanoparticles (see for example, *Nat. Mater.*, **2006**, 5, 265; *Acc. Chem. Res.*, **2008**, 41, 1662; *Nat. Chem.*, **2013**, 5, 466; *Nat. Commun.*, **2015**, 6, 6990). In addition, particles spontaneously to the air/liquid interface to minimize the interfacial energy. We have also experimentally demonstrated the decreases in surface tension upon nanoparticle adsorption to the air/liquid interface, which supports the fact that nanoparticles do adsorb to the air/liquid interface in our system to minimize the interfacial energy. As discussed earlier, it is reasonable to assume that the various Ag polyhedra in our work remain fully solvated at the

air/liquid interface during self-assembly. The associated increase in free energy for our nanoparticles to be fully exposed to air will be high, since their edge lengths are upwards of 100 nm. Simulations have shown that nanoparticles at the air/liquid interface remain completely solvated during self-assembly, and that fully exposing a ~ 30 nm nanoparticle to air requires a higher free energy input of $\sim 10^4 k_B T$ (*Nano Lett.*, **2014**, 14, 1032). Furthermore, for nanocubes system, a prior study of PbSe nanocubes using small angle X-ray scattering shows that the nanocubes do indeed adopt a [111] termination at the air/liquid interface during self-assembly to minimize exposure to air (*Nano Lett.*, **2012**, 12, 4791). At the same time, this orientation corresponds to the vertices of nanocubes, which are regions of large curvatures. These regions typically have lower surface coverages of capping ligands, and have been extensively studied in systems such as Au nanorods (*Chem. Soc. Rev.*, **2008**, 37, 1783; *Chem. Mater.*, **2013**, 25, 1250). Collectively, the primary orientation observed corresponds to one which enables maximal solvation of the capping ligands at the air/liquid interface.

Based on the findings of these works, together with our SEM observations of the nanocubes, it is rational to assume that in our system, semi-crystalline structures form at the air/liquid interface. In our case, the comprehensive order analyses discussed earlier do indeed support our earlier findings of forming two distinct structures in our system. The liquid crystal formed using octahedra building blocks, as well as the plastic crystals formed using nanocubes, truncated nanocubes, cuboctahedra, and truncated octahedra, is distinct from the corresponding densest-packed bulk supercrystals.

To clarify our discussion, we have edited the discussion in the main text:

Page 8, Paragraph 1:

Moreover, the dominant nanoparticle orientations in the mesophases correspond to the nanoparticle vertices or edges (Fig. 2e), and are regions of relatively larger curvatures as compared to the facets for the respective nanoparticle morphologies. These regions have lower densities of PVP as compared to the other parts of the nanoparticles, similar to other systems such as oleic acid-stabilized PbSe nanocubes, and oleylamine-stabilized ZnS hexagonal bipyramids/bifrustums. These prior studies show that it is thermodynamically favorable for

nanoparticles to remain solvated within the liquid phase during self-assembly, because exposing the nanoparticles to the air/liquid interface require significant additional energy input. Consequently, the main particle orientation observed in our mesophases enables an optimal amount of PVP as well as the entire particle to remain solvated within the liquid phase at the air/liquid interface during self-assembly.'

- I'm not sure the SERS data is really necessary for this paper but if the authors want to include it I would like to see some more thorough explanation for why they observe a higher enhancement factor for the plastic crystal surface compared to a more typical bulk crystal. Yes they will have higher field enhancement at tip-to-tip junctions but fewer molecules will be able to experience this larger field. In the face-to-face junctions of the simple cubic lattice the field intensity may be lower but more molecules will experience the enhancement volume. Also, how does their measurement account for differences in the number of molecules being probed in the two experiments? I would somehow need to be convinced that the same number of molecules is being probed in both supercrystals in order to conclude that the reason for the higher signal in one is because of higher field enhancement.

We understand the reviewer's concern with the fairness of comparison for the SERS enhancement factor calculations across different metacrystal structures. Our approach of using SERS enhancement factor takes into account both nanoparticle density and number of molecules probed within a laser excitation volume, and then normalized against signals obtained in the absence of metallic nanostructures (normal Raman signal). As documented in our supplementary information in our previous submission, the equation used is SERS enhancement factor = $\frac{I_{SERS}}{N_{SERS}} \div \frac{I_{ref}}{N_{ref}}$. I_{SERS} and I_{ref} refer to the SERS intensity and normal Raman intensity of 4-MBT within the laser excitation volume; N_{SERS} and N_{ref} correspond to the number of 4-MBT molecules probed during SERS and normal Raman measurements. In all cases, x-y SERS imaging measurements are carried out to obtain at least 100 spots with the area of each spot spanning $4 \mu\text{m}^2$.

We first derive the nanoparticle density per unit area for the respective metacrystals by manually examining various SEM images of the samples used for SERS experiments. We check an

average of three SEM images per metacrystal structure, and the area examined per image is $\sim 27 \mu\text{m}^2$. It is important to note that our chosen probe molecule, 4-methylbenzenethiol (4-MBT), forms a close-packed self-assembled monolayer on the nanoparticle surfaces via the formation of Ag-S bond (*Chem. Rev.*, **2005**, 105, 1103). The ligand exchange reaction is allowed to proceed for more than 48 hours, hence it is reasonable for 4-MBT to form a close-packed and well-formed monolayer on the nanoparticle surfaces. Given that a monolayer requires 4.5×10^{14} 4-MBT molecules/cm² (*Chem. Rev.*, **2005**, 105, 1103), we can estimate the number of 4-MBT molecules excited within the laser excitation volume. Note that we do not distinguish the location of the 4-MBT molecules on the nanocubes surfaces, edges, or tips. Signals collected from our samples are ensemble signals generated from the supercrystals within the laser spot.

Subsequently, we can estimate the SERS enhancement factor by comparing the SERS signals against the normal Raman signals of 4-MBT in the absence of Ag nanocubes, and normalizing the signal intensity against the number of molecules excited. This approach ensures a fair comparison of the SERS enhancement factors across different metacrystal structures, and is widely used in the estimation of SERS enhancement factors.

Figure R1-8. (a) SEM images of the (i) dual-structure nanocube supercrystal, (ii) close-packed bulk supercrystal of nanocubes, (iii) dual-structure octahedra supercrystal, and the (iv) densest-packed Minkowski lattice. (b) Corresponding SERS spectra and (c) enhancement factors of 4-MBT from the respective assembled structures. Blue and red colors correspond to the dual-structure and single bulk supercrystals respectively. Error bars in (c) are standard deviations of at least 100 measurements over areas of $4 \mu\text{m}^2$ each.

We have further tested the SERS capabilities of octahedra-based metacrystals, comparing the SERS efficiency of the dual-structure supercrystal against that of the bulk Minkowski lattice. As shown in Figure R1-8, the dual-structure supercrystal also exhibits stronger SERS signals of (962 ± 67) counts/s relative to the (366 ± 19) counts/s of the Minkowski lattice (Figure R1-8b). The resulting SERS enhancement factors are estimated to be 5.2×10^6 and 1.6×10^6 for the dual-structure supercrystal and Minkowski lattice respectively (Figure R1-8c). This observation suggests that dual-structure supercrystals are capable of enhancing the overall SERS capabilities as compared to a uniform supercrystal.

. Yes they will have higher field enhancement at tip-to-tip junctions but fewer molecules will be able to experience this larger field. In the face-to-face junctions of the simple cubic lattice the field intensity may be lower but more molecules will experience the enhancement volume.

In addition, SERS enhancement is a highly inhomogeneous phenomenon due to the inhomogeneous distribution of intense and highly localized electromagnetic fields (Physics Today, 2011, 64, 39). It was reported that 63 in one million (i.e. 0.0063%) of the total surface area contributed to 24% of the overall SERS intensity (Science 2008, 321, 388). In fact, similar observation was reported by our previous study on interfacial liquid-liquid assembly of various nanocube monolayer structures (Nano Letters 2016, 16, 3872). Our results demonstrate that hexagonal nanocubes aligned in edge-to-edge configuration with the lowest packing density of 24% generates nearly 26-fold and 350-fold stronger electromagnetic field and surface-enhanced Raman scattering as compared to the planar simple cubic nanocube structure with 100% packing density (in face-to-face configuration). This highlights that highest SERS enhancement are predominantly determined by the highly delocalized electromagnetic field. In our case (and also

shown in other reports), largest hotspot areas are observed in the lowest-packing density hexagonal metacrystal with the lowest number of particles per unit area.

We have updated Figure 4 in the revised manuscript using Figure R1-8, and have included the discussion of Ag octahedra dual-structures in the main text as:

Page 9, Second Paragraph:

‘With the ability to spontaneously organize nanoparticles into two distinct structures over large areas, we demonstrate that such structures boost the overall surface-enhanced Raman scattering (SERS) efficiency of the entire supercrystal. We use nanocubes and octahedra to compare the SERS efficiency between the dual-structure supercrystals and the densest-packed bulk supercrystal (Fig. 4a). Using 4-methylbenzenthioi (4-MBT) as the probe molecule, signature vibrational modes at 1080 cm^{-1} and 1600 cm^{-1} are observed from the SERS spectra (Fig. 4b, Table S4). The dual-structure assembly of nanocubes clearly exhibits stronger SERS intensity of (440 ± 17) counts/s relative to the simple cubic supercrystal’s (214 ± 18) counts/s. Similarly, the dual-structure assembly formed using octahedra generates higher signals of (962 ± 67) counts/s as compared to the (366 ± 19) counts/s of the Minkowski lattice. Indeed, the SERS enhancement factor (EF) for the 1080 cm^{-1} mode of the nanocube dual-structure is estimated to be 2.5×10^6 , nearly 4-fold higher than the simple cubic supercrystal (Fig. 4c). Likewise, the estimated SERS EF for the octahedra dual-structure is 5.2×10^6 , and is ~ 3.3 -fold higher than the Minkowski lattice. For nanocubes, the higher SERS EF likely arises from the stronger electromagnetic field enhancements among the hexagonal array of nanocubes than the simple cubic structure at the laser excitation wavelength⁷. Notably, this higher SERS EF arises despite the significantly lower packing density of 33 % for the topmost layer of the dual-structure assembly as compared to the simple cubic supercrystal’s 100 %. Our results collectively highlight the synergistic effect of integrating two assembled structures into one supercrystal for enhanced optical sensitivity via a particle-efficient approach.’

Overall I think this is a very nice paper and has some fantastic data, particularly that which is summarized in Figure 1. For the octahedra, it is clear that the system adopts different bulk and surface structures. However, I’m somewhat skeptical that this is also true for the other Ag

nanoparticle shapes. I'm afraid that many of the crystals were simply made in a way that they have a defective surface. I am also somewhat concerned about the strength of the data in support of the formation mechanism involving mesophases as the air-liquid interface. Because I believe the manuscript is potentially interesting I would like to give the authors a chance to address these concerns. However, if my interpretation of their data is correct then I do not see the novelty being appropriate for Nature Communications.

We are grateful for the reviewer's highly positive and constructive feedback on this manuscript, which has enabled us to look deeper into our data and refine the discussion in our revised manuscript.

To clarify the dual-structure nature of our supercrystals, we have performed comprehensive image analyses with the addition of radial distribution function (RDF) evaluations and orientational analyses using the deuterium order parameters (S_{CD}) for all nanoparticle morphologies. These additional analyses together lead us to ascertain the dual-structure nature of our supercrystals. We further hypothesize that the disruption observed between the topmost layer and the bulk supercrystal is dependent on particle adsorption at the interface and the van der Waals attraction between neighboring particles. The interplay of these forces gives rise to plastic crystals for nanocubes (smaller change in interfacial adsorption energy, higher mobility, weaker van der Waals) and liquid crystals for octahedra (larger change in interfacial adsorption energy, lower mobility, stronger van der Waals). Our combined observations here thus support the claim that the topmost layer is not simply a defective surface.

Our experimental observations focus on pioneering the formation of various dual-structure supercrystals using a family of shape-controlled Ag polyhedra, and should be of interest to the general community of materials science research. Regrettably, currently available experimental techniques are unable to shed more comprehensive insights on the formation mechanism of our dual-structure supercrystals, as well as the roles of each micro-environment in our experimental setup. This will certainly be a worthwhile area of study in future scientific research. Theoretical studies which model the interfacial behavior of nanoparticles including free-energy calculations and density functional theory can be employed to gain further insights on our system. These

calculations can also be combined with Monte Carlo simulations to attempt to decipher the roles of the individual micro-environments in our setup. This will be an exciting endeavor of its own.

In terms of material properties, we have clarified that the SERS enhancement factors comparison is fair and normalized against the number of probe molecules excited in the dual-structure supercrystal and that of a uniform supercrystal. We have also supplemented the material property investigation with the octahedral system, which further demonstrates the advantages of having a dual-structure supercrystal to generate a stronger SERS enhancement than a uniform supercrystal.

Reviewer #2 (Remarks to the Author):

Creating Two Self-Assembly Micro-Environments to Achieve Supercrystals with Dual Structures Using Polyhedral Nanoparticles.

The authors show nice self-assembly of Ag NP building blocks in different shapes: nanocubes, truncated cubes, cuboctahedra, truncated octahedra and octahedra. They use two different techniques to assemble the Ag NPs: one is drop casting on solid substrate and the second is by liquid-air interface. The main work was to compare between the top layer of the assembly and the inner part of the crystals. Moreover, the authors claim that the inner assembly parts correspond to Minkowski lattices.

The paper is written well, and the SEM images are quite nice, however, at this stage, I would expect to do more analysis in order to get deep understanding on such crystals. The paper/work is not unique and the authors did not show any special properties or phenomenon of this crystals, necessary to publish in Nature family. There are several similar works that already published, for example: 1. Wang, H.; Lee, H.-W.; Deng, Y.; Lu, Z.; Hsu, P.-C.; Liu, Y.; Lin, D.; Cui, Y. *Nat. Commun.* 2015, 6, 7261.

We thank the reviewer for the comments provided, but there are several factual errors that we would like to clarify over here.

First, to the best of our knowledge, our work on using *one self-assembly technique* for the formation of metacrystals with dual structures, namely the coexistence of the densest-packed supercrystals with their mesophases, for a family of Ag polyhedra has not been reported prior to our work. This is in contrast with the reviewer's description of our current work as not unique and not showing any special properties or phenomenon of the crystals. Previous works on the self-assembly of Ag polyhedra has resulted only in the formation of uniform densest-packed supercrystals (*Nat. Mater.*, **2012**, 11, 131). The diverse mesophases observed in this work is also unprecedented. While literature has demonstrated that PbSe nanocubes form surface structures different from the bulk, it is unclear how nanoparticle morphology affects mesophase formation

for other related polyhedra (*Nano Lett.*, **2012**, 12, 4791). Our work here represents a more complete finding for the family of highly symmetric Ag polyhedra, which gives rise to morphology-dependent mesophases on the topmost layer.

Second, as we schematically depict our experimental setup in Scheme 1 in the manuscript (Figure R2-1), we use a *single* experimental technique to assemble various Ag polyhedra. This is different from the reviewer describing our current work as using two different techniques to assemble Ag polyhedra: one by drop casting on a solid substrate and the second by the liquid-air interface. More specifically, a dispersion of the Ag polyhedra is drop cast on a substrate and allowed to dry over time. Within this drying droplet, two micro-environments are present, which lead to the formation of two distinct building block morphology-dependent metacrystal structures. The first micro-environment is near the drying front, where nanoparticle accumulation arising from solvent convective flow during drying gives rise to the densest-packed supercrystals. The second micro-environment is at the air-liquid interface, where nanoparticles adsorb to the interface to lower the interfacial energy. Subsequent recombination of the nanoparticles from both micro-environments gives rise to metacrystals with two distinct structures.

Figure R2-1. Schematic illustration of our self-assembly setup.

Third, we have performed additional analyses according to the constructive feedback provided by the other reviewers. We added new image analyses to determine the orientational order of the various mesophases and compared them against their densest-packed supercrystals. To analyze the orientational ordering of the metacrystals, we use the deuterium order parameter (S_{CD}). S_{CD} is typically used to evaluate the orientational order of hydrocarbon tails in phospholipid bilayers (*Eur. Biophys. J.*, **2007**, 36, 919). S_{CD} is calculated using the equation $S = (3\cos^2\theta - 1)/2$. A value

of $S = 1$ indicates perfect nanoparticle orientation along the crystal direction whereas $S = 0.5$ corresponds to a nanoparticle orientation perpendicular to the crystal direction. $S = 0$ indicates a random nanoparticle orientation with respect to the crystal direction. θ refers to the angle between the nanoparticle edge vector and the main crystal direction. For our system, we compare the percentage difference in S values ($\% \Delta S_{CD}$) obtained for the topmost layer with that obtained for the bulk supercrystal to correlate the relative loss in orientational order for the topmost layer. To adapt S_{CD} for our nanoparticle-based system, we manually label the nanoparticles in the SEM images to extract the cartesian coordinates. The coordinates are then used to derive one-dimensional edge vectors; these edge vectors represent the orientation of the nanoparticle edge relative to a pre-defined cartesian axis. In our analysis, we choose the main crystal direction of the assembled structure as the default axis. By comparing the edge vectors with the main crystal direction, we can obtain an angle θ to describe the nanoparticle orientation with respect to the main crystal direction.

For the Ag octahedra system, the topmost layer is a liquid crystal. The RDF profiles examining the translational ordering of the topmost layer and bulk supercrystal are shown in Figure 1d and f of the manuscript respectively (Figure R2-2). The topmost layer has lower translational order than the bulk supercrystal, as the reviewer pointed out from his observations. The lower translational order is evident from our RDF analysis as the topmost layer has lesser peaks in its RDF profile than the bulk supercrystal. Peaks are also sharper and more defined in the bulk supercrystal than the topmost layer. In the bulk supercrystal, the separation distances between neighboring particles are highly defined and regular, giving rise to discrete and intense peaks in the RDF profile (*J. Chem. Phys.* **2004**, 120, 9383). In contrast, nanoparticles in the topmost layer are displaced from the ideal lattice positions, thus generating broader and less intense peaks in its RDF profile. The broader peaks arise from a distribution of separation distances from the ideal separation. Furthermore, we note that octahedra in the topmost layer are aligned in the same direction, analogous to the smectic phases of liquid crystals observed in Au nanorods (*Angew. Chem. Int. Ed.*, **2008**, 47, 9685).

Figure R2-2. Dual crystal structure formed using Ag octahedra in water. (a, b) Cross-sectional panoramic characterization shows the formation of large-area supercrystals with long-range order. The heights of the supercrystal are nearly uniform at $\sim 10 \mu\text{m}$ and stretching more than $50 \mu\text{m}$ across. (c-h) The supercrystal can be divided into two distinct regions. (c, d) The top layer features a liquid crystal with Ag octahedra possessing long-range orientational order without translational order. (e, f) The bulk of the supercrystal is crystalline, and can be indexed to the Minkowski lattice (inset of e). Insets of d, f are the FFT images of the respective crystal structure. The Ag octahedra in the liquid crystalline phases are color-coded for ease of discussion. (g) Radial distribution function of the respective crystal.

For the orientational order analysis of the octahedra system, $S = 0.9602$ and $S = 0.9565$ for the bulk supercrystal and topmost layer respectively (Table R2-1). The orientational ordering of both the topmost layer and bulk supercrystal differs by only 0.4 %, indicating negligible loss of orientational order in the octahedra system. In addition, the standard deviation of S values for both the topmost layer and the bulk supercrystal are in the same range of $\sim 10 \%$, indicating similar distribution of orientational orderness for both structures. Together with the reduced

translational order, and the negligible loss of orientational order, we classify the topmost layer of the Ag octahedra as a liquid crystalline phase.

Table R2-1. Orientational order analysis of the various assembled structures using the respective building blocks.

	Truncated			Truncated	
	Cubes	Cubes	Cuboctahedra	Octahedra	Octahedra
S_{CD} (Bulk)	0.9806	0.8937	0.9820	0.9027	0.9602
S_{CD} (Mesophase)	0.8042	0.8003	0.7446	0.7948	0.9565
$\% \Delta S_{CD}$	18	11	24	12	0.4

We also perform the orientational order analysis for the other nanoparticle morphologies, as shown in Table R2-1. Notably, the orientational order is significantly lower for the topmost layer as compared to the corresponding bulk supercrystal for nanocubes, truncated nanocubes, cuboctahedra, and truncated octahedra. The percentage difference in S_{CD} between the bulk supercrystal and the topmost layer differs by at least 11 % and this percentage certainly contrasts with that observed for octahedral nanoparticles. The lower orientational order of the topmost layer for nanocubes, truncated nanocubes, cuboctahedra, and truncated octahedra thus supports our conclusion that the mesophases are indeed plastic crystals with distinct structures from their bulk supercrystals.

Fourth, with regards to material properties, we have also demonstrated that the metacrystals with dual structures exhibit stronger SERS capabilities than a uniform supercrystal. We have further tested the SERS capabilities of octahedra-based metacrystals, comparing the SERS efficiency of the dual-structure supercrystal against that of the uniform Minkowski lattice. As shown in Figure R2-3, the liquid crystal also exhibits stronger SERS signals of (962 ± 67) counts/s relative to the (366 ± 19) counts/s of the Minkowski lattice (Figure R2-3b). The resulting SERS enhancement factors are estimated to be 5.2×10^6 and 1.6×10^6 for the liquid crystal and Minkowski lattice respectively (Figure R2-3c). Note that our estimation for the enhancement factors has taken into account the number of probe molecules excited in both structures to ensure a fair comparison of

the enhancement factors. This observation suggests that dual-structure supercrystals are capable of enhancing the overall SERS capabilities as compared to a uniform supercrystal.

Figure R2-3. (a) SEM images of the (i) dual-structure nanocube supercrystal, (ii) close-packed bulk supercrystal of nanocubes, (iii) dual-structure octahedra supercrystal, and the (iv) densest-packed Minkowski lattice. (b) Corresponding SERS spectra and (c) enhancement factors of 4-MBT from the respective assembled structures. Blue and red colors correspond to the dual-structure and single bulk supercrystals respectively. Error bars in (c) are standard deviations of at least 100 measurements over areas of $4 \mu\text{m}^2$ each.

Fifth, we have performed comprehensive analysis before indexing the bulk supercrystal of Ag octahedra to the Minkowski lattice, which seems to be doubted by the reviewer. Specifically, the fast-Fourier transform (FFT) image of the bulk supercrystal obtained experimentally is compared against the FFT image of a model Minkowski lattice constructed using computational software (Figure R2-4). The comparison yields identical diffraction spots for both the experimental supercrystal and the model Minkowski, thereby affirming the lattice of the experimental

supercrystal as the Minkowski lattice. Figure R2-4 was in fact included in the supplementary information of our original submission as Supplementary Figure 2. This method of comparison has also been utilized in an earlier publication on the densest-packing of Ag polyhedra (*Nat. Mater.* **2012**, 11, 131).

Figure R2-4. Comparison of the (a) experimentally obtained supercrystal and its (b) FFT image with the (c) computationally generated standard Minkowski lattice and its (d) FFT image.

Sixth, the reviewer mentioned that several similar works have already been published, and cited a work published in Nature Communications as an example of a similar work. The work cited is titled ‘Bi-functional non-noble metal oxide nanoparticle electrocatalysts through lithium-induced conversion for overall water splitting’. The novelty and focus of the work is not on nanoparticle self-assembly, but on developing ultrasmall (2-5 nm) transition metal oxide nanoparticles which can simultaneously catalyze oxygen reduction reaction and hydrogen evolution reaction in a water splitting setup. On the other hand, our work here focuses on the formation of supercrystals with dual structures in a single experimental self-assembly setup. As we mentioned earlier, such supercrystals also exhibit stronger SERS capabilities as compared to a bulk supercrystal with uniform structures. Consequently, the focus of the cited reference and our current work is entirely different, including materials used, properties demonstrated, and potential applications

of the platforms. A recent related publication in Nature Communications on nanoparticle self-assembly is titled ‘Shape-dependent ordering of gold nanocrystals into large-scale superlattices’ (*Nat. Commun.*, **2017**, 8, 14038). The focus of this work is elucidating how nanoparticle shape affects the ability to assemble large-area superlattices using various gold nanocrystals with dimensions less than 100 nm. This work is again different from our current work, including the building blocks used and the potential application of our metacrystals, in which we demonstrate the outstanding SERS capabilities of the dual-structure supercrystal as compared to the uniform supercrystals.

Overall, we believe we have clarified all the doubts raised by the reviewer’s misunderstanding of our current work. We have also strengthened our data analyses in the revised manuscript, to demonstrate the dual-structure nature of our supercrystals. We have further demonstrated that on top of nanocubes, Ag octahedra-based dual-structure supercrystals also generate stronger SERS enhancements than its corresponding uniform supercrystals.

We have revised the manuscript with additional discussion on both translational and orientational order analyses. New SERS experiments have also been updated. The following changes have been made:

Page 5, Paragraph 1:

‘In contrast, octahedra in the topmost layer organize into a liquid crystal, with reduced translational order (Fig. 1 b-d, blue colored layer).’

‘...FFT analysis of the topmost layer also demonstrates the crystallinity of this structure (insets of Fig. 1d, f), with the experimentally determined packing efficiency estimated to be 49 % (Fig. S4). However, a displacement in the lattice positions of the octahedra indicates a loss of translational order.’

‘On the other hand, long-range orientational order is evident for this top layer, with octahedra oriented along the same direction. Orientational order analysis using the deuterium order

parameter indicates that there is negligible reduction of orientational order between the topmost layer and the Minkowski lattice (Table S1).'

Page 5, Paragraph 2:

'Long-range translational order of the structures is evident from the SEM, FFT, and RDF profiles for all the plastic crystals observed (Fig. 2a-d, panel ii, Fig. S6).

'The reduction in orientational order among these topmost layers is further affirmed through orientational order analyses performed using the deuterium order parameter (Table S1), thus indicating that the topmost layers are plastic crystals.'

Page 6, Paragraph 2:

'The differences observed from the FFT patterns, RDF profiles, as well as close-up SEM images for the plastic crystals and the bulk supercrystals further show that they are two distinct structures (Fig. S8).'

Page 9, Second Paragraph:

'With the ability to spontaneously organize nanoparticles into two distinct structures over large areas, we demonstrate that such structures boost the overall surface-enhanced Raman scattering (SERS) efficiency of the entire supercrystal. We use nanocubes and octahedra to compare the SERS efficiency between the dual-structure supercrystals and the densest-packed bulk supercrystal (Fig. 4a). Using 4-methylbenzenthionol (4-MBT) as the probe molecule, signature vibrational modes at 1080 cm^{-1} and 1600 cm^{-1} are observed from the SERS spectra (Fig. 4b, Table S4). The dual-structure assembly of nanocubes clearly exhibits stronger SERS intensity of (440 ± 17) counts/s relative to the simple cubic supercrystal's (214 ± 18) counts/s. Similarly, the dual-structure assembly formed using octahedra generates higher signals of (962 ± 67) counts/s as compared to the (366 ± 19) counts/s of the Minkowski lattice. Indeed, the SERS enhancement factor (EF) for the 1080 cm^{-1} mode of the nanocube dual-structure is estimated to be 2.5×10^6 , nearly 4-fold higher than the simple cubic supercrystal (Fig. 4c). Likewise, the estimated SERS EF for the octahedra dual-structure is 5.2×10^6 , and is ~ 3.3 -fold higher than the Minkowski lattice. For nanocubes, the higher SERS EF likely arises from the stronger electromagnetic field

enhancements among the hexagonal array of nanocubes **than the simple cubic structure** at the laser excitation wavelength⁷. Notably, this higher SERS EF arises despite the significantly lower packing density of 33 % **for the topmost layer of the dual-structure assembly** as compared to the simple cubic supercrystal's 100 %. Our results collectively highlight the synergistic effect of integrating two assembled structures into one supercrystal for enhanced optical sensitivity via a particle-efficient approach.'

In addition, Figure 2 has been updated to include the FFT for the bulk supercrystals, Figure 4 has been updated using Figure R2-3, Table R2-1 has been included as Table S1 (Page 5 of Supplementary Information).

The following definitions have also been supplemented with Table S1 (Page 5 of Supplementary Information):

'Nanoparticles can undergo various pathways and exhibit diverse phase behaviors during the transition from the isotropic fluid state to the crystalline state. In the isotropic state, nanoparticles are randomly dispersed within a solvent (experimentally) or within a simulation box (theoretically). Mesophases, in the form of plastic or liquid crystals, can arise during this transition from isotropic fluid state to the crystalline state. In plastic crystals, nanoparticles lose their orientational order while maintaining translational order. On the other hand, nanoparticles in liquid crystals lose their translational order while maintaining orientational order. It should be noted that this loss in order is relative, and can range from a slight loss in order, to diminished and reduced order.'

To analyze the orientational ordering of the metacrystals, we perform additional image analyses using the deuterium order parameter (S_{CD}). S_{CD} is typically used to evaluate the orientational order of hydrocarbon tails in phospholipid bilayers and is calculated using the equation $S = (3\cos^2\theta - 1)/2$. A value of $S = 1$ indicates perfect nanoparticle orientation along the crystal direction whereas $S = 0.5$ corresponds to a nanoparticle orientation perpendicular to the crystal direction. $S = 0$ indicates a random nanoparticle orientation with respect to the crystal direction. θ refers to the angle between the nanoparticle edge vector and the main crystal direction.

To adapt S_{CD} for our nanoparticle-based system, we manually label the nanoparticles in the SEM images to extract the cartesian coordinates. The coordinates are then used to derive one-dimensional edge vectors; these edge vectors represent the orientation of the nanoparticle edge relative to a pre-defined cartesian axis. In our analysis, we choose the main crystal direction of the assembled structure as the default axis. By comparing the edge vectors with the main crystal direction, we can obtain an angle θ to describe the nanoparticle orientation with respect to the main crystal direction. We compare the percentage difference in S values ($\% \Delta S_{CD}$) obtained for the topmost layer with that obtained for the bulk supercrystal to correlate the relative loss in orientational order for the topmost layer.'

- I recommend to synchronize between figure 2 and the main text (or to split figure 2). The figure is very busy and the reader loses the principle during reading.

We thank the reviewer for the constructive feedback, and have since refined the figure labeling as well as text discussion in our revised manuscript. Figure labels now appear in sequence. However, we have decided to keep the entire figure as it currently is. This is because the purpose of Figure 2 is to highlight the formation of various building block morphology-dependent plastic crystals which are entirely different from their bulk supercrystals. In addition to SEM images, we believe that additional analyses in terms of dominant particle orientation (Fig. 2e) and packing densities (Fig. 2f) enrich the information we can extract from our experimental data. Splitting the figure up will lower the completeness of the broad picture we are presenting.

- Correct the name in the main text (Fig. 4 instead Fig.5)

We apologize for this oversight, and have since amended the figure labeling.

Reviewer #3 (Remarks to the Author):

The authors present a detailed experimental study showing that Ag nanopolyhedra can be assembled into dual structures, where the order of the top layer differs from the rest of the supercrystal, via simple drying of a droplet placed on a Si surface. In addition, they show that such structures can result in SERS enhancement. This results is new to me, and I expect it to be of strong interest to others in the fields and likely to motivate future work. The paper is clearly written and presented, with appropriate reference to previous literature, and sufficient methodological detail. That said, a number of claims should probably be clarified, amended, or supported with additional data.

We thank the reviewer for the generally positive comments and subsequent constructive feedback. Below is our point-by-point response to the concerns raised by the reviewer.

1) The claim that the top layer of octahedra is a liquid crystal is technically wrong. It is neither liquid, nor does it entirely lack translational order (as claimed). In Fig. S3, typical crystal defects, including grain boundaries and vacancies, are visible. Several different types of packing, with distinct lattice parameters, are also visible. The long-range translational order appears to be reduced due to this polydispersity and perhaps a lattice mismatch between the top layer and the underlying Minkowski lattice. The claim that the topmost layer is “a liquid crystalline phase of the Minkowski lattice” also makes no sense.

We stand by our claim of the topmost layer of Ag octahedra as a liquid crystal. We have also performed additional image analyses to support our claim that the topmost layer of Ag octahedra is a liquid crystal with reduced translational order while maintaining orientational order in the following discussion.

Let us first clarify the various terms used in this manuscript. Nanoparticles can undergo various pathways and exhibit diverse phase behaviors during the transition from the isotropic fluid state to the crystalline state. In the isotropic state, nanoparticles are randomly dispersed within a solvent (experimentally) or within a simulation box (theoretically). Mesophases, in the form of plastic or liquid crystals, can arise during this transition from isotropic fluid state to the

crystalline state. In plastic crystals, nanoparticles lose their orientational order while maintaining translational order. On the other hand, nanoparticles in liquid crystals lose their translational order while maintaining orientational order (Figure R3-1, *Nat. Mater.* **2011**, 10, 230; *Science*, **2012**, 337, 453). It should be noted that this loss in order is relative, and can range from a slight loss in order, to diminished and reduced order, as per the reviewer's observations.

Figure R3-1. (a) Various possible phase behaviors of nanoparticles between isotropic and crystalline states. Reprinted with permission from *Nat. Mater.* **2011**, 10, 230. Copyright 2011 Nature Publishing Group. (b) Smectic phase (liquid crystalline phase) of Au nanorods. Adapted with permission from *Chem. Mater.* **2015**, 27, 2998. Copyright 2015 American Chemical Society.

Various types of liquid and plastic crystals have been reported using micro-/nanoparticles experimentally, including nanorods (*Phys. Rev. Lett.*, **2003**, 90, 018303), nanoplates (*ACS Nano*, **2011**, 5, 8322), and various superbubble colloids (*Nat. Commun.*, **2017**, 8, 14352). Note that in all these works, *the liquid crystals are not liquid*, and in fact some possess high translational and orientational order such as the smectic phases observed in nanorods (Figure R3-1b, *Angew. Chem. Int. Ed.*, **2008**, 47, 9685, *ACS Nano*, **2012**, 6, 4137; *Chem. Mater.*, **2015**, 27, 2998).

On the other hand, the first liquid crystals are discovered among the assembled structures using cholesteric molecules. Since then the field of organic molecules-based liquid crystal research has advanced significantly. Molecular-based liquid crystals do indeed exhibit liquid-like

characteristics. However, this term has also been adapted in the field of nanoparticle self-assembly with slightly different characteristics. We highlight here that liquid crystals observed in molecular systems are distinct from those observed in nanoparticle-based systems.

To show that the topmost layer is ‘a liquid crystalline phase’, we employ two software processing techniques, namely radial distribution function (RDF) and the deuterium order parameter (S_{CD}), on our SEM images. RDF analyzes the translational ordering of the metacrystals, while S_{CD} analyzes the orientational ordering. S_{CD} is an additional analytical software used to better correlate orientational order since the previous submission.

Figure R3-2. Dual crystal structure formed using Ag octahedra in water. (a, b) Cross-sectional panoramic characterization shows the formation of large-area supercrystals with long-range order. The heights of the supercrystal are nearly uniform at $\sim 10 \mu\text{m}$ and stretching more than $50 \mu\text{m}$ across. (c-h) The supercrystal can be divided into two distinct regions. (c, d) The top layer features a liquid crystal with Ag octahedra possessing long-range orientational order without

translational order. (e, f) The bulk of the supercrystal is crystalline, and can be indexed to the Minkowski lattice (inset of e). Insets of d, f are the FFT images of the respective crystal structure. The Ag octahedra in the liquid crystalline phases are color-coded for ease of discussion. (g) Radial distribution function of the respective crystal.

For the Ag octahedra system, the topmost layer is a liquid crystal. The RDF profiles show that the topmost layer has lower translational order than the bulk supercrystal ((Figure R3-2, Figure 1d, f), as the reviewer pointed out from his observations. The lower translational order is evident from our RDF analysis as the topmost layer has lesser peaks in its RDF profile than the bulk supercrystal. Peaks are also sharper and more defined in the bulk supercrystal than the topmost layer. In the bulk supercrystal, the separation distances between neighboring particles are highly defined and regular, giving rise to discrete and intense peaks in the RDF profile (*J. Chem. Phys.* **2004**, 120, 9383). In contrast, nanoparticles in the topmost layer are displaced from the ideal lattice positions, thus generating broader and less intense peaks in its RDF profile. The broader peaks arise from a distribution of separation distances from the ideal separation. Furthermore, we note that octahedra in the topmost layer are aligned in the same direction, analogous to the smectic phases of liquid crystals observed in Au nanorods (*Angew. Chem. Int. Ed.*, **2008**, 47, 9685).

To analyze the orientational ordering of the metacrystals, we perform additional image analyses using the deuterium order parameter (S_{CD}). S_{CD} is typically used to evaluate the orientational order of hydrocarbon tails in phospholipid bilayers (*Eur. Biophys. J.*, **2007**, 36, 919). S_{CD} is calculated using the equation $S = (3\cos^2\theta - 1)/2$. A value of $S = 1$ indicates perfect nanoparticle orientation along the crystal direction whereas $S = 0.5$ corresponds to a nanoparticle orientation perpendicular to the crystal direction. $S = 0$ indicates a random nanoparticle orientation with respect to the crystal direction. θ refers to the angle between the nanoparticle edge vector and the main crystal direction (Figure R3-3, yellow arrows and red dash lines). For our system, we compare the percentage difference in S values ($\% \Delta S_{CD}$) obtained for the topmost layer with that obtained for the bulk supercrystal to correlate the relative loss in orientational order for the topmost layer.

Figure R3-3. Orientational order analyses for Ag octahedra. SEM images labeled with the nanoparticle edge vectors for the (i) topmost layer and (ii) bulk supercrystal in yellow. Yellow arrows indicate the edge vector direction. Red dash lines indicate the main orientation of the assembled structure. (iii) Orientational order for the topmost and bulk supercrystal.

To adapt S_{CD} for our nanoparticle-based system, we manually label the nanoparticles in the SEM images to extract the cartesian coordinates (Figure R3-3, marked in yellow in panels i and ii). The coordinates are then used to derive one-dimensional edge vectors; these edge vectors represent the orientation of the nanoparticle edge relative to a pre-defined cartesian axis. In our analysis, we choose the main crystal direction of the assembled structure as the default axis (Figure R1-3, red dashed lines panels i and ii). By comparing the edge vectors with the main crystal direction, we can obtain an angle θ to describe the nanoparticle orientation with respect to the main crystal direction.

For the octahedra system, $S = 0.9602$ and $S = 0.9565$ for the bulk supercrystal and topmost layer respectively (Figure R3-3). The orientational ordering of both the topmost layer and bulk supercrystal differs by only 0.4 %, indicating negligible loss of orientational order in the octahedra system. In addition, the standard deviation of S values for both the topmost layer and the bulk supercrystal are in the same range of ~ 10 %, indicating similar distribution of orientational order for both structures. Together with the reduced translational order, and the

negligible loss of orientational order, we classify the topmost layer of the Ag octahedra as a liquid crystalline phase.

Indeed, Figure R3-2c does show a clear lattice mismatch between the top layer and the underlying Minkowski lattice, and can likely contribute to the observation of a liquid crystalline phase in the topmost layer with reduced translational order. This mismatch likely arises during the merging of the topmost layer with the bulk supercrystal. The bulk supercrystal is tilted with respect to the substrate normal, arising from convective solvent flow during solvent evaporation. The van der Waals attraction between neighboring Ag octahedra in the topmost layer is strong and reduces the interfacial mobility of the Ag octahedra. Thus, as these two structures combine during solvent drying, there is a clear lattice mismatch between these two structures.

Regarding the reviewer's opinion of Figure S3 in which 'typical crystal defects, including grain boundaries and vacancies, several different types of packing with distinct lattice parameters, are visible', we would like to emphasize that the octahedra still maintains a uniform direction analogous to a nematic liquid crystalline phase. In related nanoparticle-based systems such as for nanorods, it is also common for the particles to adopt nematic, smectic-A/B phases in the same assembly (*Angew. Chem. Int. Ed.*, **2008**, 47, 9685, *ACS Nano*, **2012**, 6, 4137; *Chem. Mater.*, **2015**, 27, 2998). It is also natural to observe defects among these liquid crystals with lower crystallinity than perfect crystals. However, these defects do not imply that the phases are not liquid crystalline anymore.

To refine our discussion in our manuscript, we have made the following changes:

Page 5, Paragraph 1:

'In contrast, octahedra in the topmost layer organize into a liquid crystal, with reduced translational order (Fig. 1 b-d, blue colored layer).'

'...FFT analysis of the topmost layer also demonstrates the crystallinity of this structure (insets of Fig. 1d, f), with the experimentally determined packing efficiency estimated to be 49 % (Fig.

S4). However, a displacement in the lattice positions of the octahedra indicates a loss of translational order.'

'On the other hand, long-range orientational order is evident for this top layer, with octahedra oriented along the same direction. Orientational order analysis using the deuterium order parameter indicates that there is negligible reduction of orientational order between the topmost layer and the Minkowski lattice (Table S1).'

The following definitions have also been supplemented with Table S1 (Page 5 of Supplementary Information):

'Nanoparticles can undergo various pathways and exhibit diverse phase behaviors during the transition from the isotropic fluid state to the crystalline state. In the isotropic state, nanoparticles are randomly dispersed within a solvent (experimentally) or within a simulation box (theoretically). Mesophases, in the form of plastic or liquid crystals, can arise during this transition from isotropic fluid state to the crystalline state. In plastic crystals, nanoparticles lose their orientational order while maintaining translational order. On the other hand, nanoparticles in liquid crystals lose their translational order while maintaining orientational order. It should be noted that this loss in order is relative, and can range from a slight loss in order, to diminished and reduced order.'

To analyze the orientational ordering of the metacrystals, we perform additional image analyses using the deuterium order parameter (S_{CD}). S_{CD} is typically used to evaluate the orientational order of hydrocarbon tails in phospholipid bilayers and is calculated using the equation $S = (3\cos^2\theta - 1)/2$. A value of $S = 1$ indicates perfect nanoparticle orientation along the crystal direction whereas $S = 0.5$ corresponds to a nanoparticle orientation perpendicular to the crystal direction. $S = 0$ indicates a random nanoparticle orientation with respect to the crystal direction. θ refers to the angle between the nanoparticle edge vector and the main crystal direction.

To adapt S_{CD} for our nanoparticle-based system, we manually label the nanoparticles in the SEM images to extract the cartesian coordinates. The coordinates are then used to derive one-

dimensional edge vectors; these edge vectors represent the orientation of the nanoparticle edge relative to a pre-defined cartesian axis. In our analysis, we choose the main crystal direction of the assembled structure as the default axis. By comparing the edge vectors with the main crystal direction, we can obtain an angle θ to describe the nanoparticle orientation with respect to the main crystal direction. We compare the percentage difference in S values ($\% \Delta S_{CD}$) obtained for the topmost layer with that obtained for the bulk supercrystal to correlate the relative loss in orientational order for the topmost layer.'

2) Calling the dual structure a "single supercrystal" is also inaccurate. It consists of one ordered structure deposited on top of a different one.

We agree with the reviewer that our supercrystals are indeed one structure deposited on top of another one, which results from single process of nanoparticle self-assembly. To minimize confusion among future readers, we have renamed our concept to 'one assembly with dual structures'.

Our experimental approach in this work is solvent evaporation-driven self-assembly, in which two micro-environments give rise to dual assembled structures within the same region on the substrate. Our approach generates the well-known coffee-ring effect, in which solvent drying drives nanoparticle deposition at the drying front. This self-assembly approach has traditionally given rise to supercrystals with uniform structures on the substrates (*Angew. Chem. Int. Ed.*, **2008**, 47, 9685; *Angew. Chem. Int. Ed.*, **2010**, 49, 6760; *Chem. Soc. Rev.*, **2011**, 40, 5457; *Angew. Chem. Int. Ed.*, **2012**, 51, 1534; *Chem. Mater.*, **2014**, 26, 4882; *Nano Lett.*, **2017**, 17, 3270). However, majority of the earlier works do not perform cross-sectional analyses of the supercrystals formed via this self-assembly approach. On the other hand, our observation of unique open structures on the topmost layer across all the building blocks motivates us to examine if subsequent deeper layers of the supercrystals also exhibit similar crystal structures. Through our cross-sectional characterization of the supercrystals in our work, we elucidate the formation of two structures from a single self-assembly process.

3) The sentence “Moreover, the dominant nanoparticle orientations in the mesophases correspond to regions with large curvatures on the respective nanoparticle morphologies” needs clarification or rewriting. Looking at the results, and considering what drives colloids to fluid-fluid interfaces, it seems at least equally likely that the particles are simply oriented in such a way that they maximise the area of interface that each particle displaces, with some variation due to orientational entropy.

We agree with the reviewer that particles are likely oriented such that they maximize the area interface that each particle displace. In addition, various other factors also impact the dominant particle orientation at the interface, including one that correspond to regions with large curvatures to enable maximum solvation of the nanoparticle and its surface capping ligands at the interface.

From our SEM image analyses, the dominant orientation is the [111] vertex facing upwards for nanocubes and truncated nanocubes (Figure R3-4e); for cuboctahedra, the dominant orientation is the [111] facets facing upwards; for truncated octahedra, the dominant orientation is the [110] edges facing upwards. Nanoparticle vertices and edges are regions of large curvatures with lower surface coverages of capping ligands in comparison to their facets (*Chem. Soc. Rev.*, **2008**, 37, 1783; *Chem. Mater.*, **2013**, 25, 1250).

Orienting these large curvature regions at the interface enables maximum solvation of the nanoparticle and its surface capping ligands at the interface. Previous nanoparticle self-assembly studies have also highlighted that nanoparticles orientate at the interface to minimize exposure to air (*Nano Lett.*, **2012**, 12, 4791; *Nano Lett.*, **2014**, 14, 1032). Simulations show that nanoparticles at the air/liquid interface remain completely solvated during self-assembly, and that fully exposing a ~ 30 nm nanoparticle to air requires additional energy input of $\sim 10^4 k_B T$. The additional free energy required for our nanoparticles to be exposed to air will be even higher, since their edge lengths are larger than 100 nm. It is therefore reasonable to assume that the Ag polyhedra in our work adopt orientations to enable full solvation of the polyhedra and capping agents at the air/liquid interface during self-assembly.

As the reviewer rightly observes, Ag polyhedra possibly adopt open structures at the interface which also maximizes the amount of interface these nanoparticles can displace. This is evident from the lower packing densities observed from the topmost layer of the mesophases relative to their densest-packed bulk supercrystals. For nanocubes system, a prior study of PbSe nanocubes using small angle X-ray scattering shows that the nanocubes do indeed adopt a [111] termination at the air/liquid interface during self-assembly to minimize exposure to air (*Nano Lett.*, **2012**, 12, 4791). This orientation is similar to the orientation of nanocubes observed in our nanocube plastic crystal.

To clarify our discussion, we have added the following highlighted sentences to our discussion in the main text:

Page 8, Paragraph 1:

‘Moreover, the dominant nanoparticle orientations in the mesophases correspond to the nanoparticle vertices or edges (Fig. 2e), and are regions of relatively larger curvatures as compared to the facets for the respective nanoparticle morphologies. These regions have lower densities of PVP as compared to the other parts of the nanoparticles, similar to other systems such as oleic acid-stabilized PbSe nanocubes, and oleylamine-stabilized ZnS hexagonal bipyramids/bifrustums. These prior studies show that it is thermodynamically favorable for nanoparticles to remain solvated within the liquid phase during self-assembly, because exposing the nanoparticles to the air/liquid interface require significant additional energy input. Consequently, the main particle orientation observed in our mesophases enables an optimal amount of PVP as well as the entire particle to remain solvated within the liquid phase at the air/liquid interface during self-assembly.’

4) It is difficult to justify the argument that directional entropic forces are responsible for formation of the Minkowski lattice when there is relatively little face-to-face contact between octahedra in this structure. Indeed, adding a depletant results in an increase in face-to-face contact and a change to a less dense lattice, as demonstrated in Ref. 14.

Directional entropic forces indeed contribute to the formation of densest-packed lattices in our assembly system. In addition, we would like to emphasize that the Minkowski lattice is the densest-packed lattice structure with the most face-to-face alignment for the octahedral system. Our reasons are elaborated below.

Directional entropic forces are emergent as particles adopt local dense configurations, and arise from the shape entropy of anisotropic nanoparticles to align neighboring particles face-to-face (*Proc. Natl. Acad. Sci.*, **2014**, 111, E4812; *ACS Nano*, **2012**, 6, 609). These forces are on the order of several $k_B T$ at the onset of crystallization, and are expected to gain dominance as the system become crowded in the absence of other intrinsic forces. While we are unable to experimentally measure the magnitude of directional entropic forces in our system, several experimental observations lead us to postulate the possible contribution of these forces in driving supercrystal formation in our experiments.

Regarding the octahedral particles, we have discussed briefly in the manuscript as well as in greater detail in the supplementary text that the Minkowski lattice is the densest-packed lattice structure with the most face-to-face alignment. To achieve a packing efficiency of $\sim 94.7\%$ in this primitive triclinic lattice, the building blocks maximize face-to-face contact with neighboring particles within the unit cell in all three dimensions. Every particle in the Minkowski lattice has 14 nearest neighbors contacting each other through 3 types of face-to-face alignment (Figure S2, R3-4). The contact areas between neighboring octahedra range from $2/3$ to $2/9$ of the [111] triangular facets of the octahedra. In contrast, the next highest packing density structure for octahedra, the hexagonal close-packed lattice, has only 8 nearest neighbors. The hexagonal close-packed lattice is planar, with 6 nearest neighbors contacting 6 facets of the octahedral particle on the same plane and 1 nearest neighbor in each plane above and below the unit cell (Figure R3-4e). The packing efficiency of this lattice is $\sim 89\%$. Finally, with regards to the helical structure reported in reference 14 in the presence of a depletant, the packing efficiency of this structure is $\sim 82\%$ and with significantly reduced face-to-face alignment. Based on the tetramer motif of the helical structure shown below (Figure R3-4f), there are 7 face-to-face contact points, which accounts for its lower packing density in comparison to the other two structures. We note that depletion-induced attraction is negligible in our experiments,

because excess capping agent (poly(vinylpyrrolidone, PVP) are removed via centrifugation prior to the self-assembly experiments.

Figure R3-4. (a-d) Schematic illustration of the simplest repeat unit of the Minkowski lattice. (a) The repeat unit comprises eight octahedral particles. Among the eight octahedra, the yellow octahedron labeled 1 in the center is in contact with the remaining seven octahedra. In the full unit cell however, there are seven more octahedra surrounding octahedra 1. These seven octahedra can be categorized into three different contacting patterns based on the different contact area they share. The largest contact area covering $2/3$ of the $[111]$ triangular facets occurs between (c) octahedra 1 and 2 (red), with the facet center in direct contact with each other. The second type of contact occurs between (b) octahedra 1 and 3-5 (purple) with a contacting area of $4/9$ of the $[111]$ triangular facets. The tip of the yellow octahedron is in contact with the three purple octahedra at $1/3$ of the octahedra edge. (d) The least area of contact with $2/9$ of triangle surface arises between the octahedra 1 and 6-8 (blue), with the tips of the yellow octahedron located at the center of the $[111]$ facets of the blue octahedra. (e) The hexagonal close-packed unit cell, which is the next densest-packed lattice of octahedral particles, has six contacting neighbors. Stacking this unit cell in three dimensions gives rise to a total of eight contacting neighbors. (f) Helical structure with seven neighbors as shown in ref. 14. Reprinted with permission from *Nat. Mater.* **2012**, 11, 131. Copyright 2012 Nature Publishing Group.

The fact that all particle morphologies organize into their densest-packed lattices point to the fact that particles adopt hard-particle behaviors during self-assembly, since particle morphology alone determines their crystal structures (*Nat. Mater.* **2012**, 11, 131). This also implies that directional entropic forces could contribute to driving supercrystal formation in our experiments.

Moreover, nanoparticles of other shapes all exhibit significant face-to-face contact in their densest-packed supercrystals. A notable example here would be the cuboctahedra supercrystal, where we observe the exclusive formation of body-centered tetragonal lattice, in which one cuboctahedron is in contact with 16 nearest neighbors (Figure R3-5). Out of these 16 neighbors, 12 cuboctahedra are in contact via face-to-face alignment. Given its morphological similarity with truncated cubes, these particles can potentially also be organized into a simple cubic supercrystal with 10 nearest neighbors with 6 in face-to-face alignment (*Nat. Nanotechnol.*, **2007**, 2, 435).

Figure R3-5. Various contact modes within the body-centered tetragonal lattice of cuboctahedra. (a) Body-centered tetragonal lattice. (b) Face-to-face and (c) edge-to-edge alignment within the same layer. (d) Face-to-face contact between neighboring layers. There is a total of 16 nearest neighbors for every building block in this lattice.

We have amended the discussion in the main text as follows:

Page 9:

‘Notably, a single Ag octahedron in the Minkowski lattice is in contact with 14 other nearest neighbors via three types of face-to-face alignment (Fig. S2)’

5) What does the following sentence mean and why is it justified? “At the same time, the orientation of the Ag polyhedra in the tilted bulk supercrystals coincides with the main particle orientation observed among the mesophase in the topmost layer.” Particles in the topmost layer adopt several different orientations. Do they all coincide with the orientation of polyhedra in the underlying supercrystal and, if so, how precisely?

We apologize for the confusion caused by this statement. Further analyses show that there are in fact misalignment between the main particle orientation of the plastic crystals and the bulk supercrystal, as pointed by the reviewer.

To analyze the orientational ordering of the metacrystals, we use the deuterium order parameter (S_{CD}). S_{CD} is typically used to evaluate the orientational order of hydrocarbon tails in phospholipid bilayers (*Eur. Biophys. J.*, **2007**, 36, 919). S_{CD} is calculated using the equation $S = (3\cos^2\theta - 1)/2$. A value of $S = 1$ indicates perfect nanoparticle orientation along the crystal direction whereas $S = 0.5$ corresponds to a nanoparticle orientation perpendicular to the crystal direction. $S = 0$ indicates a random nanoparticle orientation with respect to the crystal direction. θ refers to the angle between the nanoparticle edge vector and the main crystal direction. For our system, we compare the percentage difference in S values ($\% \Delta S_{CD}$) obtained for the topmost layer with that obtained for the bulk supercrystal to correlate the relative loss in orientational order for the topmost layer. To adapt S_{CD} for our nanoparticle-based system, we manually label the nanoparticles in the SEM images to extract the cartesian coordinates. The coordinates are then used to derive one-dimensional edge vectors; these edge vectors represent the orientation of the nanoparticle edge relative to a pre-defined cartesian axis. In our analysis, we choose the main crystal direction of the assembled structure as the default axis. By comparing the edge vectors with the main crystal direction, we can obtain an angle θ to describe the nanoparticle orientation with respect to the main crystal direction.

For the orientational order analysis of the octahedra system, $S = 0.9602$ and $S = 0.9565$ for the bulk supercrystal and topmost layer respectively (Table R3-1). The orientational ordering of both the topmost layer and bulk supercrystal differs by only 0.4 %, indicating negligible loss of orientational order in the octahedra system. In addition, the standard deviation of S values for

both the topmost layer and the bulk supercrystal are in the same range of $\sim 10\%$, indicating similar distribution of orientational orderness for both structures. Together with the reduced translational order, and the negligible loss of orientational order, we classify the topmost layer of the Ag octahedra as a liquid crystalline phase.

Table R3-1. Orientational order analysis of the various assembled structures using the respective building blocks.

	Truncated			Truncated	
	Cubes	Cubes	Cuboctahedra	Octahedra	Octahedra
$S_{CD} (Bulk)$	0.9806	0.8937	0.9820	0.9027	0.9602
$S_{CD} (Mesophase)$	0.8042	0.8003	0.7446	0.7948	0.9565
$\% \Delta S_{CD}$	18	11	24	12	0.4

We also perform the orientational order analysis for the other nanoparticle morphologies, as shown in Table R3-1. Notably, the orientational order is significantly lower for the topmost layer as compared to the corresponding bulk supercrystal for nanocubes, truncated nanocubes, cuboctahedra, and truncated octahedra. The percentage difference in S_{CD} between the bulk supercrystal and the topmost layer differs by at least 11% and this percentage certainly contrasts with that observed for octahedral nanoparticles. The lower orientational order of the topmost layer for nanocubes, truncated nanocubes, cuboctahedra, and truncated octahedra thus supports our conclusion that the mesophases are indeed plastic crystals with distinct structures from their bulk supercrystals.

Furthermore, a closer examination of the assembly with dual-structure show that there is a misalignment is between the topmost layer and the bulk supercrystal (Figure R3-6). Among the various nanoparticle morphologies, the smallest misalignment is $\sim 5^\circ$ for nanocubes.

Figure R3-6. Close-up cross-sectional SEM images of (a) nanocubes, (b) truncated nanocubes, (c) cuboctahedra, (d) truncated octahedra, and (e) octahedra indicating the disruption between the topmost layer of nanoparticles (highlighted in blue) and the bulk supercrystal.

We hypothesize that the competition between nanoparticle interfacial adsorption behavior and the subsequent recombination with the bulk supercrystal during solvent drying contributes to the extent of disruption observed between the topmost layer and the bulk supercrystal, and hence the clarity of the dual-structure supercrystal. This hypothesis arises from a qualitative trend observed in Figure R3-6: the disruption between the topmost layers and the bulk layers becomes increasingly evident as the particle morphology transits from nanocubes to octahedra. Our hypothesis is based on several main considerations that we will elaborate in detail in subsequent paragraphs. First, combined SEM observation with existing literature findings points to the fact that nanoparticles at the interface do indeed form a semi-crystalline structure. Secondly, the

interfacial mobility of nanoparticles as well as the van der Waals attraction between nanoparticles at the interface is dependent on particle sizes. The combination of these factors thus give rise to orientational disorder among the nanocube plastic crystal and translational disorder among the octahedra liquid crystal as the topmost layer merges with the bulk supercrystal during drying.

The original discussion has been removed from the revised manuscript, Table R3-1 has been included as Table S1 (Page 5 of Supplementary Information), and Figure R3-6 has been included as Figure S8 (Page 9 of Supplementary Information).

6) The arguments that dual structures lead to SERS enhancement in general and that this is due to synergistic effects are not adequately supported by the data. In particular, it is unclear whether shapes other than cubes will give rise to SERS enhancement, and it is unclear whether the enhancement is simply due to the spikier top layer rather than a property of the dual structure.

The higher SERS enhancement factor of the dual-structure supercrystal arises from the overall larger-area local electromagnetic fields generated by the plastic crystal structure and the densest-packed bulk metacrystal.

We have further tested the SERS capabilities of octahedra-based metacrystals, comparing the SERS efficiency of the dual-structure supercrystal against that of the bulk Minkowski lattice. Note that the octahedral topmost layer does not possess “spiky structures” as that of the nanocube, effectively eliminating the contribution of “lightning rod” effect from the octahedra dual structure’s SERS performance. As shown in Figure R3-10, the dual-structure supercrystal also exhibits stronger SERS signals of (962 ± 67) counts/s relative to the (366 ± 19) counts/s of the Minkowski lattice (Figure R3-7b). The resulting SERS enhancement factors are estimated to be 5.2×10^6 and 1.6×10^6 for the dual-structure supercrystal and Minkowski lattice respectively (Figure R3-7c). This observation suggests that dual-structure supercrystals are capable of enhancing the overall SERS capabilities as compared to a uniform supercrystal.

Figure R3-7. (a) SEM images of the (i) dual-structure nanocube supercrystal, (ii) close-packed bulk supercrystal of nanocubes, (iii) dual-structure octahedra supercrystal, and the (iv) densest-packed Minkowski lattice. (b) Corresponding SERS spectra and (c) enhancement factors of 4-MBT from the respective assembled structures. Blue and red colors correspond to the dual-structure and single bulk supercrystals respectively. Error bars in (c) are standard deviations of at least 100 measurements over areas of 4 μm^2 each.

We have updated Figure 4 in the revised manuscript using Figure R3-7, and have included the discussion of Ag octahedra dual-structures in the main text as:

Page 9, Second Paragraph:

‘With the ability to spontaneously organize nanoparticles into two distinct structures over large areas, we demonstrate that such structures boost the overall surface-enhanced Raman scattering (SERS) efficiency of the entire supercrystal. We use nanocubes and octahedra to compare the SERS efficiency between the dual-structure supercrystals and the densest-packed bulk supercrystal (Fig. 4a). Using 4-methylbenzenthionol (4-MBT) as the probe molecule, signature

vibrational modes at 1080 cm^{-1} and 1600 cm^{-1} are observed from the SERS spectra (Fig. 4b, Table S4). The dual-structure assembly of nanocubes clearly exhibits stronger SERS intensity of (440 ± 17) counts/s relative to the simple cubic supercrystal's (214 ± 18) counts/s. Similarly, the dual-structure assembly formed using octahedra generates higher signals of (962 ± 67) counts/s as compared to the (366 ± 19) counts/s of the Minkowski lattice. Indeed, the SERS enhancement factor (EF) for the 1080 cm^{-1} mode of the nanocube dual-structure is estimated to be 2.5×10^6 , nearly 4-fold higher than the simple cubic supercrystal (Fig. 4c). Likewise, the estimated SERS EF for the octahedra dual-structure is 5.2×10^6 , and is ~ 3.3 -fold higher than the Minkowski lattice. For nanocubes, the higher SERS EF likely arises from the stronger electromagnetic field enhancements among the hexagonal array of nanocubes than the simple cubic structure at the laser excitation wavelength⁷. Notably, this higher SERS EF arises despite the significantly lower packing density of 33 % for the topmost layer of the dual-structure assembly as compared to the simple cubic supercrystal's 100 %. Our results collectively highlight the synergistic effect of integrating two assembled structures into one supercrystal for enhanced optical sensitivity via a particle-efficient approach.'

Reviewers' comments:

Reviewer #1 (Remarks to the Author):

I very much appreciate the detailed response provided by the authors to my comments. They have clearly considered all points raised and addressed them. Below are my specific comments to those points that I believe are in greatest disagreement:

- I have a much better understanding of the authors' use of terms "translational order", "orientational order", "plastic crystal", "liquid crystal", etc... I am fully convinced that the original claim is correct, i.e. that the topmost layer of octahedra has reduced translational order but long-range orientational order. Many thanks to the authors for the additional orientational correlation analysis to convince me of this. Although I don't wish to be pedantic by focusing so heavily on the semantics of their text, I unfortunately still have some trouble with how the results of Figure 1 are described. As stated by myself and the authors, while the translational order of the topmost layer is reduced, it is not absent. However, the caption of Figure 1 says, "possessing long-range orientational order without translational order" (emphasis mine). Separately, the authors state, "displacement in the lattice positions of the octahedra indicates a loss of translational order." Translational order isn't lost, it is reduced. I feel that a statement like this is both inaccurate and needlessly confusing for the reader.

- I am somewhat troubled by the use of the terms "plastic crystal" and "liquid crystal" in this manuscript. I am quite familiar with the definitions the authors have used of lacking translational or orientational order to define liquid and plastic crystals respectively. But is this all that is necessary to call something a liquid or plastic crystal? I'm not so sure. In the references the authors have provided (Nat. Mater. 2011, 10, 230; Science, 2012, 337, 453), these phases arise from entropic crowding effects when particles are mobile in a solvent. The authors describe the formation of plastic or liquid mesophases as arising from the gradual transition from isotropic to crystalline phases. But this is not how the surface mesophase is forming in the present case. Here, particles are merely sedimenting onto a template defined by the corrugations of the underlying bulk crystal, and then drying out. It feels as if the forces driving the appearance of this surface mesophase are quite different. If I were to lithographically define a template that captured and oriented particles on a surface (by using specific ligand chemistry for example) and then removed all the solvent, would this be a liquid or plastic crystal based on how I defined my template? Somehow I don't think so. The mechanism of formation is totally different. Simply the absence of either translational or orientational order seems to me to be a necessary but not sufficient requirement. Calling the octahedra mesophases observed here liquid crystals or plastic crystals still seems inaccurate or misleading to me. I remain uncomfortable with the use of these terms and strong language like "loss of translational order". It feels very overstated to me, which really isn't necessary since I think these results are novel and interesting on their own.

- Moving on to the set of data discussing the cubes, cuboctahedra, and truncated octahedra, I still do not understand how one can claim that the surfaces are totally different crystal structures from the bulk. Simply having some disorder (orientational or otherwise) at a surface, does not constitute an entirely distinct phase. In atomic systems, nearly all surfaces experience some disordering relative to the bulk (e.g. lattice parameter and strain relaxation). But we do not consider all surfaces to be entirely different crystals. Even surface reconstructions are not generally considered an entirely new phase. Perhaps this disagreement comes back to our conflicting definitions of plastic crystal and the entropic origins of the ordering but I am still failing to see how the surfaces of these nanocrystal superlattices are not merely a bit defective, as most surfaces are.

- The observation of hexagonal vs square packing in SEM images of the topmost surface and bulk crystal, respectively, is totally unconvincing. These are merely the different perspectives one sees

when viewing a square lattice of cubes along the [111] and [100] directions, respectively. See the short video I attached. The first few frames show a {111} surface of a simple cubic arrangement of cubes viewed along the [111] direction. The perspective then changes to show the same crystal along the [100] direction. One orientation shows a hexagonal arrangement and the other a square arrangement. But these are exactly the same crystal. In this case it is clear then that the RDF would show peaks in different positions for these different crystal perspectives. Seeing different diffraction spots is not, in and of itself, indicative of a crystal with different symmetry. In addition, the topmost layer is just more defective so the peaks are broadened and less numerous. Similar claims apply to the cuboctahedra and octahedra results. The arguments that make reference to other cubic nanostructures orienting along the [111] at air/liquid interfaces also don't do much for me as these effects will be highly-dependent on the particular surfactants present at the surface, the particle composition, the surface ligands on the particle, the particle size, etc... It may be true that the same [111] orientation is adopted in this case but I can't see a reason why this would be reasonable to assume without evidence. Overall, these data leave me only more convinced of my original explanation, rather than that of the authors. I'm afraid I still see this as a big problem as it seriously impacts the novelty of the work.

- I would also like to note that the radial distribution function for the topmost layer of octahedra (R1-2g) indicates considerably longer-range order than that for the topmost layer of cubes (R1-5a). However, the authors describe the octahedra as "without translational order" while they describe the cubes as having "long-range translational order". This is merely to emphasize my point about confusing language and claims that appear a little over-reaching in furtherance of a narrative.

- The authors provide quite reasonable arguments for the assembly mechanism based on air-liquid interfacial interactions. But I simply don't see much evidence for these proposed mechanisms. Some thorough characterization of the particle orientation at these air-liquid interfaces would be hugely helpful in bolstering their claims. I understand that instrument limitations might make this difficult but I can't ignore the lack of supporting data for this claim. The authors might be correct, but as it currently stands, I believe my proposed explanation of a defective surface is just as plausible an explanation for Figure 2 as what the authors have provided.

Although the paper has been improved since the previous review, many of the major issues I had remain. I still feel as if the language is confusing, several claims are over-reaching, and some of the crucial data is being misinterpreted. I remain steadfast in my claim that the authors have some really interesting results here. But at this point I have to recommend publication in a different journal.

Reviewer #2 (Remarks to the Author):

I have previously review this manuscript for Nature communication. In this new version, the authors have addressed all the reviewers concerns and, therefore, the manuscript should be considered for publication.

Reviewer #3 (Remarks to the Author):

In my opinion, most of the points raised in the previous round of review have been satisfactorily addressed by the authors in their extensive, if somewhat repetitive, reply. For example, I'm convinced that the surface layers are structurally distinct from the underlying bulk crystals in that they have reduced translational or orientational order, even if they are not liquid or plastic crystals in the traditional sense. And while it is not entirely clear to what extent the surface order is determined by the way that the particles order at the air-liquid interface vs the way that they interact with the exposed bulk crystal surface, I do not think that it is essential to answer this question in the present work.

On the question of directional entropic forces, this term is commonly used to refer to the tendency of faceted particles to align with pairs of facets overlapping, either in the presence of depletant molecules (e.g. DOI: 10.1002/anie.201306009) or upon an increase in particle packing fraction (e.g. cubes). If the authors want to use the term simply to mean that hard anisotropic particles order orientationally as well as translationally upon densification, then they should be aware that it may cause some misunderstanding among readers.

Reviewer #1 (Remarks to the Author):

I very much appreciate the detailed response provided by the authors to my comments. They have clearly considered all points raised and addressed them. Below are my specific comments to those points that I believe are in greatest disagreement:

1. I have a much better understanding of the authors' use of terms "translational order", "orientational order", "plastic crystal", "liquid crystal", etc... I am fully convinced that the original claim is correct, i.e. that the topmost layer of octahedra has reduced translational order but long-range orientational order. Many thanks to the authors for the additional orientational correlation analysis to convince me of this. Although I don't wish to be pedantic by focusing so heavily on the semantics of their text, I unfortunately still have some trouble with how the results of Figure 1 are described. As stated by myself and the authors, while the translational order of the topmost layer is reduced, it is not absent. However, the caption of Figure 1 says, "possessing long-range orientational order without translational order" (emphasis mine). Separately, the authors state, "displacement in the lattice positions of the octahedra indicates a loss of translational order." Translational order isn't lost, it is reduced. I feel that a statement like this is both inaccurate and needlessly confusing for the reader.

We apologize for the oversight in the wording of our revised manuscript. As we have stated in our reply earlier, the liquid and plastic crystals observed in our work have reduced translational/orientational order relative to their bulk supercrystals. We have tidied up the relevant description of the decrease in translational/orientational order of the liquid/plastic crystals in the latest manuscript.

The sentence describing the radial distribution function of Ag octahedra in the main text now reads as 'A displacement in the lattice positions of the octahedra in this topmost layer indicates a decrease in translational order relative to the bulk Minkowski lattice. Order analyses performed using radial distribution function also show broader, less distinct, and lower-intensity peaks with increasing particle separation for the topmost octahedra layer, affirming decreased translational order of this layer relative to the Minkowski lattice.' Figure 1's caption which states '...without translational order...' has also been corrected to 'The top layer features a liquid crystal with Ag octahedra possessing long-range orientational order with reduced translational order relative to the bulk.'

In addition, the description in the second paragraph of the introduction has also been adjusted to ‘In our experiments, these polyhedra spontaneously organize into large-area structurally diverse densest-packed supercrystals as well as their liquid (reduced translational order relative to the bulk) and plastic (reduced orientational order relative to the bulk) crystals.’

2. I am somewhat troubled by the use of the terms “plastic crystal” and “liquid crystal” in this manuscript. I am quite familiar with the definitions the authors have used of lacking translational or orientational order to define liquid and plastic crystals respectively. But is this all that is necessary to call something a liquid or plastic crystal? I’m not so sure. In the references the authors have provided (Nat. Mater. 2011, 10, 230; Science, 2012, 337, 453), these phases arise from entropic crowding effects when particles are mobile in a solvent. The authors describe the formation of plastic or liquid mesophases as arising from the gradual transition from isotropic to crystalline phases.

With regards to the definition of liquid/plastic crystals, we have reviewed various literature and conclude that these terms are predominantly used to describe the physical appearance of the assembled structures, including both molecular and nanoparticle systems. However, we emphasize that how these assembled structures arise is often not considered in the first instance in the reported literature.

The overarching definition for liquid and plastic crystals relates back to the respective decrease/loss of translational or orientational order observed among assembled structures. Liquid crystals include nematic, smectic-A, smectic-C, hexatic and discotic phases, all of which exhibit high orientational order with various extents of translational order (Principles of Condensed Matter Physics, 1995, Cambridge University Press). Notably, the authors in this book further highlight that the origin of molecular nematic/smectic phases arises from the threadlike structures in the experimental observations of G. Friedal. The subsequent naming of lyotropic or thermotropic liquid crystals also refers to liquid crystals which undergo phase transitions as a function of solvent or temperature changes. Timmermans coined the term plastic crystals, in which the rotational/orientational loss in certain molecular plastic crystals significantly reduces the mechanical strengths of such solids (*J. Phys. Chem. Solids* **1961**, 18, 1). Therefore, the terms for liquid and plastic crystals arises as a consequence of physical appearance or physical properties, and the mechanisms giving rise to the formation of these assembled structures are not always known in the first instance.

Similarly, various liquid and plastic crystals have been continuously observed in nanoparticle-based systems. Even though many systems can generate liquid or plastic crystals at intermediate volume fractions during the isotropic to crystal transition (such as the references mentioned by the reviewer), there are also systems for which the formation mechanism for liquid/plastic crystal remains unresolved to date. For liquid crystal systems, we direct the reviewer to several references in which nanoplates and nanorods are shown to organize into liquid crystalline structures without fully elucidating their formation mechanisms (Figure R1-1, *Angew. Chem. Int. Ed.* **2008**, 47, 9685; *ACS Nano* **2011**, 5, 8322). The Au nanorods shown here are assembled using droplet evaporation, in which the presence of surfactants is required for liquid crystal formation (*Angew. Chem. Int. Ed.* **2008**, 47, 9685). The authors suggested that the surfactants are required to balance the entropic depletion potential and electrostatic repulsion potential. On the other hand, liquid crystals of GdF₃ nanoplates are formed via an interfacial self-assembly approach (*ACS Nano* **2011**, 5, 8322). Organic dispersions of the nanoplates are spread over the surface of a polar solvent and subsequently allowed to dry. The authors postulated that hydrophobic interactions with the subphase influences the types of liquid crystals observed from the nanoplates. In both cases, the formation of liquid crystals is not fully understood. For plastic crystal systems, colloidal rods are recently shown to give rise to plastic crystals (*Nat. Commun.* **2014**, 5, 3092). While the authors demonstrate the existence of plastic crystalline phase over a large range of volume fractions as well as the ability to use electric fields to modulate rod orientations, the driving force behind plastic crystal formation is not explicitly discussed.

Figure R1-1. Liquid crystals assembled using (a, b) Au nanorods and (c-d) GdF₃ nanoplates. Reprinted with permission from *Angew. Chem. Int. Ed.* **2008**, 47, 9685 and *ACS Nano* **2011**, 5, 8322. Copyright 2008 Wiley-VCH Verlag GmbH & Co. and 2011 American Chemical Society.

Consequently, we believe that the naming or categorization of assembled structures as liquid or plastic crystals should only be directly correlated with the physical appearance or the physical properties of the assembled structures. The formation mechanism can arise from a multitude of factors aside from entropic ordering, especially in systems involving shape-controlled nanoparticles. Interactions between shape-controlled nanoparticles as well as the interactions of these nanoparticles with their immediate environment (including the presence of an interface) are significantly more complex than a simple spherical particle.

But this is not how the surface mesophase is forming in the present case. Here, particles are merely sedimenting onto a template defined by the corrugations of the underlying bulk crystal, and then drying out. It feels as if the forces driving the appearance of this surface mesophase are quite different.

We demonstrate here that our surface liquid crystal is not formed by ‘merely sedimenting onto a template defined by the corrugations of the underlying bulk crystals and then drying out’. We perform an additional experiment to establish the importance of the air/liquid interface in providing a second self-assembly micro-environment for the formation of liquid crystals using Ag octahedra. This experiment involves trapping the Ag octahedra in-situ at the air/liquid interface during self-assembly, allowing us to visualize the orientation of the Ag octahedra at the air/liquid interface during self-assembly prior to complete solvent drying. Such visualization will enable us to separate the influence of the interface from the role of the bulk supercrystal in driving liquid crystal formation on the topmost layer. The experimental approach involves the formation of a thin polymer film, poly(ethylcyanoacrylate) (PECA), at the air/liquid interface (*Nanoscale* **2014**, 6, 6879). Ethyl cyanoacrylate monomers are introduced to the droplet of Ag octahedra dispersion via the gas phase in a closed environment. Anionic polymerization of ethyl cyanoacrylate occurs upon contact with the aqueous droplet surface, and is initiated through a nucleophilic attack by water molecules on the ethyl cyanoacrylate monomers. The polymer film grows from the air/liquid interface towards the interior of the water droplet. In our setup, we allow the self-assembly process to proceed for an hour before introducing the ethyl cyanoacrylate monomers for polymerization. The PECA film is then subjected to the usual SEM characterization.

SEM characterization clearly shows the formation of a semi-crystalline monolayer at the air/liquid interface, with emerging translational and orientational order among the Ag octahedra in as short as one hour into the self-assembly experiment (Figure R1-2). We further note that the orientation of these ‘frozen’ Ag octahedra bear striking resemblance to the eventual liquid crystal that we observe in Figure 1 of the manuscript. We observe qualitatively lower translational and orientational order among the Ag octahedra as compared to the eventual liquid crystal formed after complete solvent drying. However, this is expected since the assembly was frozen just one hour into the drying process, and that PECA polymerization continued as the solvent vaporized. Since PECA polymerization occurs from the droplet surface towards the interior of the droplet, we can also characterize the extent of particle exposure to the air on the droplet surface. Closer-up examination of the Ag octahedra shows that only the triangular [111] facets of the particles are slightly exposed above the PECA film, translating to a similar exposure at the air/liquid interface (Figure R1-2c). Majority of the Ag octahedra remain solvated within the droplet. This finding is consistent with both literature our earlier discussion on particles adopting configurations which enable maximal solvation of both nanoparticles and the capping agent during self-assembly.

Figure R1-2. Visualizing the organization of Ag octahedra at the air/liquid interface via the polymerization of ethyl cyanoacrylate at (a) low and (b) higher magnifications. A semi-crystalline monolayer form just one hour into self-assembly with emerging translational order. (c) Majority of the Ag octahedra remain embedded within the polymer film to minimize exposure to air.

The similarity of this semi-crystalline monolayer with the liquid crystal formed after solvent drying leads us to conclude that the air/liquid interface indeed creates a second micro-environment in our assembly to enable liquid crystal formation in the dual-structure supercrystal. In fact, our findings here corroborate well with an earlier work (*Phys. Rev. Lett.* **2004**, 93, 135503), which utilizes in-situ small angle X-ray scattering to examine the dynamic self-assembly of spherical Au nanoparticles (~ 8 nm) during colloidal droplet evaporation. This work shows that the Au nanoparticles can organize

into crystalline superlattices at the interface. Consequently, it should not be surprising to find that the use of shape-controlled nanoparticles can also lead to liquid crystal formation at the air/liquid interface, and that the liquid crystal does not arise from a mere particle sedimentation onto a corrugated surface. Since the liquid crystal formation in our system is closely related to the air/liquid interface and does not simply arise during the isotropic to crystalline transition, we have removed the use of ‘mesophase’ in our manuscript discussion and in this reply. We hope this will minimize confusion on our discussion, because in simulations, mesophase implies ‘order-disorder transition involving changes in both translational and orientational degrees of freedom’ (*Nat. Mater.* **2011**, 10, 230).

If I were to lithographically define a template that captured and oriented particles on a surface (by using specific ligand chemistry for example) and then removed all the solvent, would this be a liquid or plastic crystal based on how I defined my template? Somehow I don’t think so. The mechanism of formation is totally different. Simply the absence of either translational or orientational order seems to me to be a necessary but not sufficient requirement. Calling the octahedra mesophases observed here liquid crystals or plastic crystals still seems inaccurate or misleading to me.

We emphasize here the focus of this current work is demonstrating the ability to create dual-structure supercrystals using two self-assembly micro-environments. Template-assisted nanoparticle self-assembly is an entirely different research area in the broad field of nanoparticle self-assembly. The primary focus of template-assisted nanoparticle self-assembly is to utilize well-defined templates to create regularly spaced single nanoparticles or small nanoparticle clusters, followed by an investigation on their material properties (see for example *Nat. Nanotechnol.* **2007**, 2, 570; *Soft Matter* **2009**, 5, 1129; *Proc. Natl. Acad. Sci.* **2013**, 110, 6640). Because every repeat pattern in the template is used to create identical organization of single nanoparticles/nanoparticle clusters, less attention is placed on their naming. Whether such templated approaches can be used to achieved liquid/plastic crystals, or dual-structure supercrystals is not relevant to our current work.

As discussed at length earlier, the categorization of liquid or plastic crystals is independent of the formation mechanism. Even well-aligned particles continue to be labeled as liquid crystals (Figure R1-1).

I remain uncomfortable with the use of these terms and strong language like “loss of translational order”. It feels very overstated to me, which really isn’t necessary since I think these results are novel and interesting on their own.

We again apologize for our oversight with regards to the use of ‘strong’ language. As mentioned earlier, the ‘loss’ of translational order has been changed to reduced translational order.

3. Moving on to the set of data discussing the cubes, cuboctahedra, and truncated octahedra, I still do not understand how one can claim that the surfaces are totally different crystal structures from the bulk. Simply having some disorder (orientational or otherwise) at a surface, does not constitute an entirely distinct phase. In atomic systems, nearly all surfaces experience some disordering relative to the bulk (e.g. lattice parameter and strain relaxation). But we do not consider all surfaces to be entirely different crystals. Even surface reconstructions are not generally considered an entirely new phase. Perhaps this disagreement comes back to our conflicting definitions of plastic crystal and the entropic origins of the ordering but I am still failing to see how the surfaces of these nanocrystal superlattices are not merely a bit defective, as most surfaces are.

As shown in our previous reply, we have adopted the standard analytical approaches to determine both translational and orientational order in our dual-structure supercrystals, using radial distribution function profiles and deuterium order parameter respectively. Our analyses clearly highlight the difference in the topmost and bulk supercrystals for the polyhedra. We emphasize again over here that the naming of plastic or liquid crystals should be based primarily on the physical appearance or properties of the assembled structures. After all, this was how the terms nematic, smectic phases come about. Subsequent use of hexagonal- or disc-like systems gave rise to the use of hexatic and discotic liquid crystalline phases. In a similar thread, the liquid and plastic crystals observed in our work are the first to be reported for shape-controlled nanoparticles.

Further to our discussion in Figure R1-2, we demonstrate here that Ag nanocubes also form a semi-crystalline monolayer at the air/liquid interface in the same in-situ trapping experiment (Figure R1-3). Nanocubes clearly assemble into a hexagonal structure at the air/liquid interface, notably with majority of the nanocubes aligned with their [111] tips facing upwards. Again, this ‘frozen’ assembled structure resembles the eventual dried state in Figure 2 of our manuscript. As with the dried state, some cubes are rotated in this structure (Figure R1-3b). Since the surface functionality of

all Ag polyhedra used in our work is the same, we can reasonably extrapolate our findings based to the other building block morphologies used in our work.

Figure R1-3. Visualizing the organization of Ag nanocubes at the air/liquid interface via the polymerization of ethyl cyanoacrylate at (a) low and (b) higher magnifications.

While we have established that nanoparticles adsorb to the interface driven by the minimization of interfacial free energy, the underlying driving force giving rise to particle morphology-dependent assembled structures remains an open question. One recent work simulating the interfacial deformation of nanocubes provides some insight in this aspect (*Phys. Rev. Lett.* **2016**, 116, 258001). The authors demonstrate that single nanocubes generate a hexapolar capillary deformation upon adsorption to an interface, and the 3-fold symmetry of such hexapoles further give rise to hexagonal or graphene-like honeycomb lattices among assembled nanocubes. Remarkably, the hexagonal lattice of assembled nanocubes observed in the simulations also bears a strikingly resemblance to the assembled nanocube structures in our work. The finding from the simulations exemplifies the complexity associated in understanding the interfacial behaviors of shape-controlled polyhedral particles. The formation of plastic crystals in our work can arise from thermal fluctuations of the solvent contact line during solvent drying. In addition, slight vertex rounding of the nanocubes can also contribute to decreased orientational order (nanocube rotation) at the air/liquid interface (*Proc. Natl. Acad. Sci.* **2011**, 108, 2684). This postulation is also supported by our experimental observations for nanocubes and truncated nanocubes. A greater population of truncated nanocubes is rotated from the dominant [111] orientation, and the orientational order of truncated nanocubes in the topmost layer is lower than that of the nanocubes.

In addition, we do not think it is a fair comparison between atomic systems and our assembly system using shape-controlled Ag polyhedra. Atomic systems are primarily characterized by covalent or ionic interactions between neighboring atoms. Strain relaxation is characterized by a change in

interlayer spacing between the top layer of atoms and the bulk layers, while surface reconstruction involves displacement of surface atoms to reduce the surface dangling bonds. It should be noted that the topmost layer remains congruent with the bulk lattices in these scenarios. On the other hand, the topmost layers observed in our shape-controlled nanoparticles represent a distinct break between the topmost layer and the bulk supercrystal, and is most obvious in the octahedra system (Figure 1C, Figure S8). Moreover, nanoparticle systems involve significantly more complex interactions, including van der Waals forces, electrostatic interactions, magnetic interactions, molecular surface interactions, and entropy (*Small* **2009**, 5, 1600). The use of shape-controlled nanoparticles together with a dynamic self-assembly environment such as a drying droplet only serves to increase the intrinsic complexity of the nanoparticle interactions in our current system. Furthermore, particle morphology also directly impacts the resulting supercrystal lattice as well as the capillary deformation at the interface. Given the vast disparity between atomic and shape-controlled nanoparticle systems, we do not think there is a good and fair basis to compare these two entities.

4. The observation of hexagonal vs square packing in SEM images of the topmost surface and bulk crystal, respectively, is totally unconvincing. These are merely the different perspectives one sees when viewing a square lattice of cubes along the [111] and [100] directions, respectively. See the short video I attached. The first few frames show a {111} surface of a simple cubic arrangement of cubes viewed along the [111] direction. The perspective then changes to show the same crystal along the [100] direction. One orientation shows a hexagonal arrangement and the other a square arrangement. But these are exactly the same crystal. In this case it is clear then that the RDF would show peaks in different positions for these different crystal perspectives. Seeing different diffraction spots is not, in and of itself, indicative of a crystal with different symmetry. In addition, the topmost layer is just more defective so the peaks are broadened and less numerous. Similar claims apply to the cuboctahedra and octahedra results.

We would like to point out that the reviewer has misunderstood the orientation of the nanocubes in our supercrystals in the provided video. We have extracted the respective images from the reviewer's video in Figure R1-4a below. For ease of comparison and discussion, we assume nanocubes to have edge lengths of 100 nm (similar to experimental dimensions). To achieve the same crystal structure with different projections as the reviewer claims, there are huge gaps of 100 nm between neighboring cubes (equivalent to the size of one nanocube, turquoise hexagon in Figure R1-4a). 'Neighboring' cubes are in a face-to-face alignment.

Figure R1-4. (a) Images of nanocubes extracted in sequence from the reviewer's video. (b, c) Top view SEM images of supercrystals assembled using nanocubes in our current work. (d) Side view SEM image of the same nanocube supercrystal. Turquoise square and hexagons highlight the repeat units in the reviewer's and our system. (e) Reconstruction of (a -iv) by removing the gaps among nanoparticles to form bulk close-packed nanocubes. In the bulk, neighboring nanocubes contact each other directly face-to-face.

However, there are no such gaps and contact modes present in our nanocube supercrystal (Figure R1-4b-e). As shown in the hexagonal repeat unit, neighboring nanocubes contact each other edge-to-edge (Figure R1-4b), and there is no gap present between neighboring nanocubes (Figure R1-4c). The side view SEM image in Figure R1-4d further illustrates the different packing habits of the topmost layer and the bulk. The top surface is undulating, arising from the predominant [111] orientation of the nanocubes in this layer. Occasional nanocubes are also observed to deviate from this [111] orientation, which contributes to the observed decrease in orientational order of this topmost layer. On the other hand, the bulk supercrystal features close-packed nanocubes, with the nanocubes contacting each other via face-to-face. It should be noted that there is no gap present between neighboring nanocubes in our assembled structures.

To further prove our point, we perform a “reverse engineering” on the reviewer’s scheme (Figure R1-4a-iv). We first remove these gaps to reconstruct the reviewer’s scheme to form Figure R1-4e-iv, and rotate it to the same angle as Figure R1-4a-i to form Figure R1-4e-i. As clearly indicated in both schemes, we cannot obtain an identical structure as the one observed in our experiments (Figure R1-4b). Consequently, our assembled nanocube supercrystal is an entirely different structure as that proposed by the reviewer. While it is true that seeing different diffraction spots is not conclusive of the nanocubes adopting distinct structures, we have highlighted a clear visual analysis of the assembled structures formed using nanocubes to demonstrate the difference in the structure of the topmost layers and the bulk. Similar differences are also evident for the other systems as shown in Figure 2 of the manuscript.

The arguments that make reference to other cubic nanostructures orienting along the [111] at air/liquid interfaces also don’t do much for me as these effects will be highly-dependent on the particular surfactants present at the surface, the particle composition, the surface ligands on the particle, the particle size, etc... It may be true that the same [111] orientation is adopted in this case but I can’t see a reason why this would be reasonable to assume without evidence.

We have maintained our original argument, in which nanocubes and other polyhedra adopt a semi-crystalline orientation at the air/liquid interface, as clearly evident in our latest experiment results in Figure R1-2 and Figure R1-3.

We point out that the reviewer has misread our discussion over here in relation to the similarity of nanocube structures observed between our work and the literature. In our work, Ag nanocubes are capped by relatively hydrophilic poly(vinylpyrrolidone) and have average edge lengths of ~ 110 nm. The original published work in which [111] orientation of nanocubes are also observed utilizes PbSe nanocubes which are capped by hydrophobic oleic acid ligands (*Nano Lett.* **2012**, 12, 4791). The average edge lengths of the PbSe nanocubes are 13.3 nm. Our experimental observations for the orientations of nanocubes are similar to this work, despite having different surfactants on the particle surface and different particle composition. Our argument here is further supported by the computational study which shows that nanocubes with homogeneous surface properties generate hexapolar capillary deformation profiles at the interface to minimize interfacial free energy (*Phys. Rev. Lett.* **2016**, 116, 258001). Such configuration leads to an equilibrium [111] orientation which further enables formation of hexagonal assembled lattices.

Overall, these data leave me only more convinced of my original explanation, rather than that of the authors. I'm afraid I still see this as a big problem as it seriously impacts the novelty of the work.

As discussed above, we have used new experimental results to demonstrate and clarify the distinction between the orientation and packing habits of the Ag polyhedra in our work from that of the reviewer's perspective.

5. I would also like to note that the radial distribution function for the topmost layer of octahedra (R1-2g) indicates considerably longer-range order than that for the topmost layer of cubes (R1-5a). However, the authors describe the octahedra as "without translational order" while they describe the cubes as having "long-range translational order". This is merely to emphasize my point about confusing language and claims that appear a little over-reaching in furtherance of a narrative.

With regards to this issue, we first clarify that we have since tidied up our discussion pertaining to the Ag octahedra system. The sentence describing the radial distribution function of Ag octahedra in the main text now reads as 'A displacement in the lattice positions of the octahedra in this topmost layer indicates a decrease in translational order relative to the bulk Minkowski lattice. Order analyses performed using radial distribution function also show broader, less distinct, and lower-intensity peaks with increasing particle separation for the topmost octahedra layer, affirming decreased translational order of this layer relative to the Minkowski lattice.' Figure 1's caption which states '...without translational order...' has also been corrected to 'The top layer features a liquid crystal with Ag octahedra possessing long-range orientational order with reduced translational order relative to the bulk.'

Next, we emphasize that radial distribution function comparisons should only be made among building blocks with the same morphology. The radial distribution functions of octahedra and nanocubes should not be compared for two reasons. Firstly, different building block morphology assemble into different supercrystals with distinct morphology-dependent packing habits. For instance, nanocubes form the simple cubic supercrystals whereas octahedra assemble into the triclinic Minkowski lattice in the bulk. Secondly, the dimensions of our building blocks are different. The edge lengths of the nanoparticles increase from ~ 110 nm in nanocubes to ~ 350 nm in octahedra. The difference in edge lengths in turn impacts the interparticle separation in the radial distribution function profiles of the respective assembled structures. Consequently, our discussion in the manuscript is based on the comparison between the topmost layers and their corresponding bulk

supercrystals for the respective building block morphology. This discussion is also evident in various published works, in which comparison of the radial distribution profiles are only carried out for single particle morphologies (see for example *Nat. Commun.* **2017**, 8, 14352; *Phys. Rev. Lett.* **2015**, 115, 078301; *J. Chem. Phys.* **2004**, 120, 9383).

6. The authors provide quite reasonable arguments for the assembly mechanism based on air-liquid interfacial interactions. But I simply don't see much evidence for these proposed mechanisms. Some thorough characterization of the particle orientation at these air-liquid interfaces would be hugely helpful in bolstering their claims. I understand that instrument limitations might make this difficult but I can't ignore the lack of supporting data for this claim. The authors might be correct, but as it currently stands, I believe my proposed explanation of a defective surface is just as plausible an explanation for Figure 2 as what the authors have provided.

We have discussed extensively, using new experimental results, in Figure R1-2 and R1-3 the importance of the air/liquid interface to support the formation of a semi-crystalline monolayer even prior to complete solvent drying.

The in-situ nanoparticle trapping observations discussed in Figure R1-2 and Figure R1-3 clearly show that Ag nanoparticles adsorb to the air/liquid interface almost 'spontaneously', forming a semi-crystalline monolayer with emerging translational and orientational order in as short as one hour. Ag nanoparticles minimize their exposure to the air. These observations lead us to conclude that the air/liquid interface indeed creates a second micro-environment in our assembly to enable the formation of a liquid crystal in the dual-structure supercrystal. Even though we used Ag octahedra and nanocubes as the model system, we believe that the findings can be extrapolated to other particle morphologies used in the study since their surface functionalities are identical.

Our findings here only serve to validate our reference to earlier published works, in which nanocubes orientate with a [111] orientation at the air/liquid interface to minimize the air/liquid interfacial energy (*Nano Lett.* **2012**, 12, 4791). While we are unable to identify the predominant driving force for the formation of the respective liquid/plastic crystals, we hypothesize that particle shape plays an important role. Just as nanocubes have been shown to create hexapolar capillary deformation which lead to the [111] interfacial orientation (*Phys. Rev. Lett.* **2016**, 116, 258001), future research can work towards elucidating the various complex particle morphology-dependent capillary deformation which can give rise to the respective assembled liquid/plastic crystals.

Although the paper has been improved since the previous review, many of the major issues I had remain. I still feel as if the language is confusing, several claims are over-reaching, and some of the crucial data is being misinterpreted. I remain steadfast in my claim that the authors have some really interesting results here. But at this point I have to recommend publication in a different journal.

We are grateful for the reviewer's efforts in ensuring accurate description, and have since taken extra care to ensure consistent language use in our latest manuscript. We have also furnished extra experiments to prove that our claims are not over-reaching. Together with extensive literature support, we are confident that our findings are now adequately substantiated.

Reviewer #2 (Remarks to the Author):

I have previously review this manuscript for Nature communication. In this new version, the authors have addressed all the reviewers concerns and, therefore, the manuscript should be considered for publication.

We thank the reviewer for the positive affirmation of our work.

Reviewer #3 (Remarks to the Author):

In my opinion, most of the points raised in the previous round of review have been satisfactorily addressed by the authors in their extensive, if somewhat repetitive, reply. For example, I'm convinced that the surface layers are structurally distinct from the underlying bulk crystals in that they have reduced translational or orientational order, even if they are not liquid or plastic crystals in the traditional sense. And while it is not entirely clear to what extent the surface order is determined by the way that the particles order at the air-liquid interface vs the way that they interact with the exposed bulk crystal surface, I do not think that it is essential to answer this question in the present work.

We thank the reviewer for the positive affirmation of our work. As we have discussed extensively in response to reviewer 1, we have some preliminary data that clearly demonstrates the importance of the air/liquid interface in directing liquid/plastic crystal formation in our dual-structure supercrystal. Future work will be conducted to more clearly elucidate the self-assembly process.

On the question of directional entropic forces, this term is commonly used to refer to the tendency of faceted particles to align with pairs of facets overlapping, either in the presence of depletant molecules (e.g. DOI: 10.1002/anie.201306009) or upon an increase in particle packing fraction (e.g. cubes). If the authors want to use the term simply to mean that hard anisotropic particles order orientationally as well as translationally upon densification, then they should be aware that it may cause some misunderstanding among readers.

We are aware the reviewer's concern over here. We have in fact included the relevant citation in our original manuscript which directs readers to the specific reference for further discussion on directional entropic forces. At the same time, we have also clarified that depletion-induced attractive forces are negligible in our experimental setup. Excess capping agents are removed via centrifugation prior to self-assembly experiments.

Reviewers' comments:

Reviewer #1 (Remarks to the Author):

I'm afraid I feel that the authors continue to use misleading language and are drawing conclusions from their data that are incorrect. I don't intend to provide much detail here since my previous responses have been quite long and clear about my concerns. The bottom line is this: If the authors include only the data from the octahedra, then I believe their claims of "two self-assembly microenvironments" are correct. Their data is convincing that at the air-liquid interface the octahedra form a different packing which then assembles onto the bulk crystal which has a different structure. I have always maintained that this is most interesting data and if this system were the only data presented in the paper, I would not have many issues. My two major problems are: (1) the description of this top phase as a liquid or plastic crystal, which I still believe to be highly misleading, (2) the remaining data on other polyhedra, from which the authors claim the effect is general is completely unconvincing. For point (1), the authors have noted that many previous papers have assigned the terms "plastic" or "liquid" crystal to systems that have assembled without complete understanding of the mechanism. This isn't very convincing to me. Most, if not all, of the cases where I've seen these terms applied have been where entropic crowding has been a reasonable expectation as participating in the assembly process. Some of the examples they outline in the response letter describe plastic phases arising at certain volume fractions of rods (i.e. from entropic crowding) – this is precisely the context I am referring to. At the very least, the particles/molecules are usually still mobile in another phase when they are in a liquid or plastic crystalline state. This is not the case in the present work. I very strongly feel that using these terms is wrong when the particles are totally static and dried on the top of the bulk crystal. My example still stands as a challenge to their terminology- dried particles assembled via a template would never be called plastic or liquid. For point (2) the authors have conveniently focused on a feature of my video that is completely inconsequential to my point- the large spacing between the cubes. Yes they are technically correct that my video did not perfectly reflect the close-packing of their system (this was to more clearly illustrate the symmetry difference). To make it more clear, I have rendered another image of the {111} facet of a simple cubic arrangement of cubes that are close-packed and compared it to their SEM image. The two are identical. I really do not understand this 'edge sharing' point. The surface they are looking at is unequivocally a disordered {111} facet of a simple cubic arrangement of cubes. It is NOT a separate crystal structure (plastic, liquid, or otherwise). This has been a point of constant disagreement and perhaps we are talking past one another but I can easily replicate the arrangement of polyhedra they see on the surface using the bulk ordering. Again, it would be grossly overinterpreting these data to say that they point to a different crystal structure on the surface. This hugely impacts their novelty as the effect they convincingly showed for the octahedra is not obviously generalizable to other polyhedra. I strongly encourage them to re-examine these data and remove the characterization of the surface as a separate crystal. Over the course of reviewing this paper I have felt more and more as if the authors are overly concerned with pushing the interpretation of their data to make the manuscript sound more interesting than is actually warranted. At this point I really don't see there being much possibility for agreement. Only if they were to completely remove the claims about liquid/plastic crystal and completely remove the data on polyhedra other than octahedra could I see myself being able to consider this paper. I feel that this is a shame since I really do like the octahedra data and I am fully convinced of the claims there. But if the authors insist that the other points stand then I cannot support publication in Nature Communications.